## RESEARCH ARTICLE

# GAK antagonises ROCK-dependent regulation of actomyosin dynamics

Masaki Hiramoto[1],*, Naoharu Takano[1], Hiroko Kokuba[2], Hiromi Kazama[1], Hirotsugu Hino[1], Hiroshi Handa[3] and Keisuke Miyazawa[1]

## ABSTRACT

Cytoskeletal proteins, such as actin and myosin, are essential for regulating cell morphology and motility. Rho-associated kinase (ROCK; herein referring collectively to ROCK1 and ROCK2) is a key regulator of actomyosin dynamics, including stress fibre formation and cell migration. Our previous study revealed that cyclin G-associated kinase (GAK) antagonises ROCK signalling during autophagic flux regulation; however, the underlying molecular mechanisms remained unclear. In this study, we investigated the role of GAK in cytoskeletal dynamics and cell motility by genetically disrupting GAK. GAK-knockout cells exhibited enhanced stress fibre formation and cell migration, accompanied by increased phosphorylation of myosin light chain (MLC). Notably, the inhibition of stress fibre formation and MLC phosphorylation was more dependent on the intrinsically disordered region (IDR) of GAK than on its kinase activity. GAK IDR interacted with ARHGEF2, and ARHGEF2 knockdown suppressed stress fibre formation in GAK-knockout cells. Furthermore, the GAK IDR contributes to the regulation of MLC expression. Together, these findings indicate that GAK IDR is a crucial regulator of actomyosin dynamics and cell motility, and suggest that GAK antagonises ROCK-dependent cytoskeletal regulation through the coordinated control of ARHGEF2 activity and MLC expression.

KEY WORDS: GAK, ROCK, ARHGEF2, Actomyosin, Stress fibre, Cell migration

## INTRODUCTION

The actin cytoskeleton plays fundamental roles in various cellular processes, including cell morphogenesis, motility, cytokinesis and vesicle trafficking (Chakrabarti et al., 2021; Clarke and Martin, 2021; Lappalainen et al., 2022). In addition to forming actin filaments through the polymerisation of globular actin proteins, these filaments assemble into contractile actomyosin structures, such as stress fibres, in conjunction with myosin filaments (Garrido-Casado et al., 2021; Vicente-Manzanares et al., 2009). Among the structural components of the cytoskeleton, stress fibres are essential for maintaining cell shape, adhesion and contractility in non-muscle cells (Burridge and

Guilluy, 2016; Livne and Geiger, 2016; Parsons et al., 2010; Pellegrin and Mellor, 2007). The formation of short branched actin networks, such as lamellipodia, generates the physical force necessary for leading-edge protrusion, whereas the contractile force generated by stress fibres drives trailing-edge retraction, enabling efficient cell migration (Hotulainen and Lappalainen, 2006; Schaks et al., 2019).

The contractile activity of myosin is regulated by the phosphorylation of myosin light chain (MLC, herein referring to the protein encoded by *MYL9*) (Vicente-Manzanares et al., 2009). A key regulator of MLC phosphorylation is myosin phosphatase, which dephosphorylates the myosin regulatory light chain, resulting in decreased myosin activity and actomyosin relaxation (Somlyo and Somlyo, 2003). Myosin phosphatase is composed of a catalytic subunit (protein phosphatase 1, PP1c, of which there are several variant in mammals) and regulatory subunits, including myosin phosphatase-targeting subunit 1 (MYPT1, also known as PPP1R12A) and myosin phosphatase Rho-interacting protein (MPRIP) (Kiss et al., 2019; Koga and Ikebe, 2005; Mulder et al., 2004; Surks et al., 2003, 2005). Myosin phosphatase activity is tightly controlled by upstream signals, particularly Rho-associated kinase (ROCK; herein referring collectively to ROCK1 and ROCK2 unless otherwise specified) signalling, which inhibits its activity by phosphorylating MYPT1 to sustain myosin contractility (Julian and Olson, 2014; Kiss et al., 2019; O'Connor and Chen, 2013).

Previous studies have highlighted the importance of Rho family GTPases and their downstream effectors, including ROCK, in the regulation of stress fibre formation and cell motility (Julian and Olson, 2014; Narumiya and Thumkeo, 2018; O'Connor and Chen, 2013). Rho-ROCK signalling promotes myosin activation by directly phosphorylating MLC and inhibiting MYPT1, and it facilitates actin polymerisation by inhibiting cofilin proteins, thereby enhancing cytoskeletal contractility and cell migration (Julian and Olson, 2014; Narumiya and Thumkeo, 2018; O'Connor and Chen, 2013). Although RhoA-mediated activation of ROCK is well characterised, accumulating evidence indicates that ROCK activity can also be modulated through Rho-independent mechanisms (Araki et al., 2001; Truebestein et al., 2015). How the upstream regulators integrate these pathways to fine-tune ROCK-dependent actomyosin dynamics remains unclear. Meanwhile, Rho GTPase activity is tightly regulated by guanine nucleotide exchange factors (RhoGEFs) and GTPase-activating proteins (Bos et al., 2007; Cherfils and Zeghouf, 2013; Patel and Karginov, 2014). RhoGEFs activate Rho proteins by promoting the exchange of GDP for GTP. One well-characterised RhoGEF is ARHGEF2 (also known as GEF-H1), whose dysregulated expression has been implicated in cancer progression and other pathological conditions involving aberrant cell motility (Fine et al., 2020; Joo and Olson, 2021). However, the regulatory mechanisms governing this pathway remain poorly characterised.

Cyclin G-associated kinase (GAK) is a serine/threonine kinase involved in clathrin-mediated endocytosis (Greener et al., 2000;

[1]Department of Biochemistry, Tokyo Medical University, Tokyo 160–8402, Japan. [2]Laboratory of Electron Microscopy, Institute of Medical Science, Tokyo Medical University, Tokyo 160–8402, Japan. [3]Molecular Pharmacology Division, Institute of Medical Science, Tokyo Medical University, Tokyo 160–8402, Japan.

*Author for correspondence (hiramoto@tokyo-med.ac.jp)

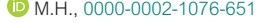 M.H., 0000-0002-1076-6515

Lee et al., 2006, 2005; Zhang et al., 2005; Zhao et al., 2001), cell cycle regulation (Fukushima et al., 2017; Naito et al., 2012; Shimizu et al., 2009; Yabuno et al., 2019) and autophagic flux (Miyazaki et al., 2021; Munson et al., 2021; Zhang et al., 2025, 2023). *GAK* has also been reported to carry single nucleotide polymorphisms (SNPs) associated with Parkinson's disease (UK Parkinson's Disease Consortium et al., 2011; Dumitriu et al., 2011; Ma et al., 2015; Pankratz et al., 2009; Rhodes et al., 2011) and to exhibit altered expression in specific cancer types, including osteosarcoma and prostate cancer (Ray et al., 2006; Sakurai et al., 2014; Susa et al., 2010). Our previous study has demonstrated that GAK antagonises ROCK signalling in regulating autophagic flux (Miyazaki et al., 2021). Given that ROCK is a key regulator of actomyosin regulation, we investigated the functional role of GAK in cytoskeletal dynamics and cell motility via targeted GAK disruption in this study.

## RESULTS

### GAK disruption induces stress fibre formation

We previously generated GAK-knockout (KO) cell lines using CRISPR/Cas9 genome editing in the human lung cancer cell line A549 and examined the role of GAK in the autophagy–lysosomal system (Miyazaki et al., 2021). *GAK* was completely knocked out in clones 1-1 and 2-1, but not in clone 1-2. Clone 1-1 was primarily used as the GAK-KO cell line for the performed analysis (Fig. S1A). That study showed that GAK KO induced morphological changes in autophagosomes and autolysosomes (Miyazaki et al., 2021). Transmission electron microscopy of GAK-KO cells also revealed the accumulation of filamentous structures, primarily at the cell periphery (Fig. 1A); however, the details had not been analysed. Phalloidin staining revealed actin filaments in the cell cortex of A549 cells, whereas prominent formation of stress fibres, which are composed of

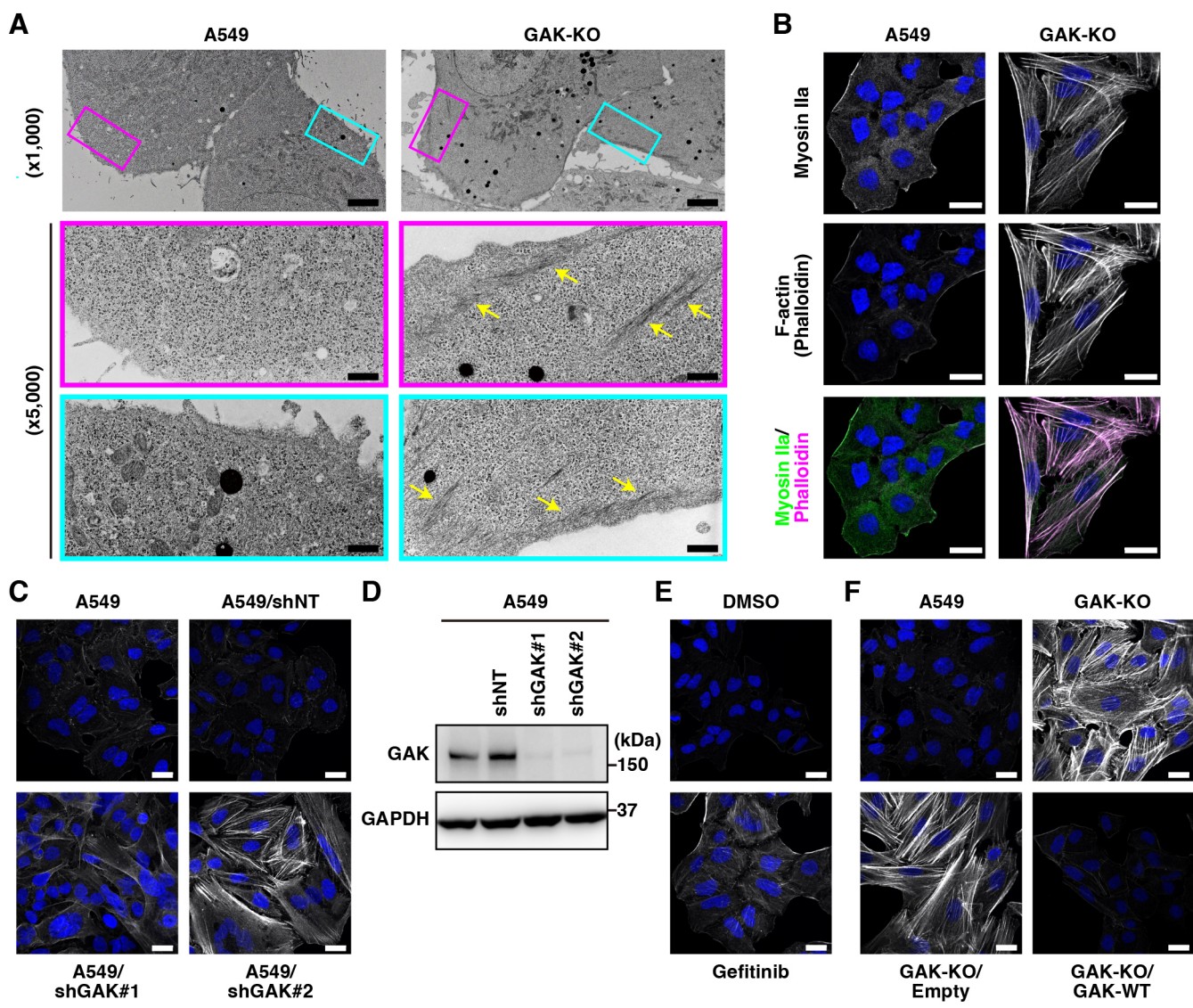

**Fig. 1. GAK disruption induces stress fibre formation.** (A) Transmission electron microscopy of wild-type (WT) A549 lung cancer cell line and the corresponding GAK-KO cells (×1000). Boxed regions are shown at high magnification (×5000). Yellow arrows indicate filamentous structures. Scale bars: 5 µm (×1000); 1 µm (×5000). (B) Immunofluorescence (IF) images of A549 WT and GAK-KO cells stained for myosin heavy chain (myosin IIa; green in merge), F-actin (phalloidin; magenta in merge) and DNA (DAPI; blue). Scale bars: 20 µm. (C) F-actin (phalloidin) staining in A549, non-targeting shRNA-transduced (A549/shNT), and GAK shRNA-transduced (A549/shGAK#1 and A549/shGAK#2) cells. Scale bars: 20 µm. (D) IB analysis of GAK expression in A549, A549/shNT, A549/shGAK#1, and A549/shGAK#2 cells. (E) F-actin (phalloidin) staining in A549 cells treated with or without gefitinib (30 µM, 3 h) treatment. Scale bars: 20 µm. (F) F-actin (phalloidin) staining in A549 WT, GAK-KO and GAK-KO cells transduced with an empty vector (GAK-KO/Empty) or GAK-WT (GAK-KO/GAK-WT). Scale bars: 20 µm. All panels show representative images from at least three independent experimental repeats.

F-actin and myosin, was observed in GAK-KO cells (Fig. 1B; Fig. S1B,C). Additionally, immunostaining of other cytoskeletal proteins, such as tubulin, keratin and vimentin, revealed only minor changes in their expression levels and intracellular distribution owing to GAK KO, in contrast to the pronounced changes observed with actin filaments (Fig. S1D). Stress fibre formation was also observed upon GAK knockdown in A549 cells (Fig. 1C,D) and similarly in the human pancreatic cancer cell line PANC-1, the human liver cancer cell line HepG2, the human oral cancer cell line CAL27 and the human neuroblastoma cell line SH-SY5Y (Fig. S1E,F). In addition, weak stress fibres were observed in A549 cells treated with gefitinib (Fig. 1E), which has been reported to inhibit the kinase activity of GAK as an off-target effect (Brehmer et al., 2005; Ohbayashi et al., 2018). Similarly, stress fibre formation was observed in A549 cells treated with specific GAK kinase inhibitors (GAK inhibitor or SGC-GAK-1) (Fig. S1G,H). Furthermore, reintroducing wild-type (WT) GAK into GAK-KO cells eliminated stress fibres (Fig. 1F). Collectively, these results indicate that GAK deficiency induces prominent stress fibre formation, suggesting that GAK plays a role in actomyosin regulation.

## GAK disruption promotes cell migration

Stress fibres are contractile actomyosin bundles that contribute to cell motility (Livne and Geiger, 2016; Pellegrin and Mellor, 2007). Therefore, we examined the effects of GAK disruption on cell migration. First, we performed a wound-healing assay. In A549 cells, ∼70% of the gap remained after 48 h, whereas in GAK-KO cells, the wound gap was nearly completely closed (Fig. 2A,B; Movies 1,2). Reintroduction of GAK-WT into GAK-KO cells restored the gap to ∼40% (Fig. 2A,B). We further conducted a random migration assay, which revealed that the relative migration distance in GAK-KO cells increased more than fourfold compared to A549 cells (Fig. 2C–E; Movies 3,4). These findings indicate that GAK disruption induces cell migration.

## GAK disruption promotes MLC phosphorylation

The contractility of stress fibres is regulated by phosphorylation of MLC (Vicente-Manzanares et al., 2009). Therefore, we examined the effects of GAK disruption on MLC phosphorylation. Expression of total MLC and phosphorylation at Thr18 and Ser19 were both increased in GAK-KO cells (Fig. 3A; Fig. S2A). An increase in Ser19

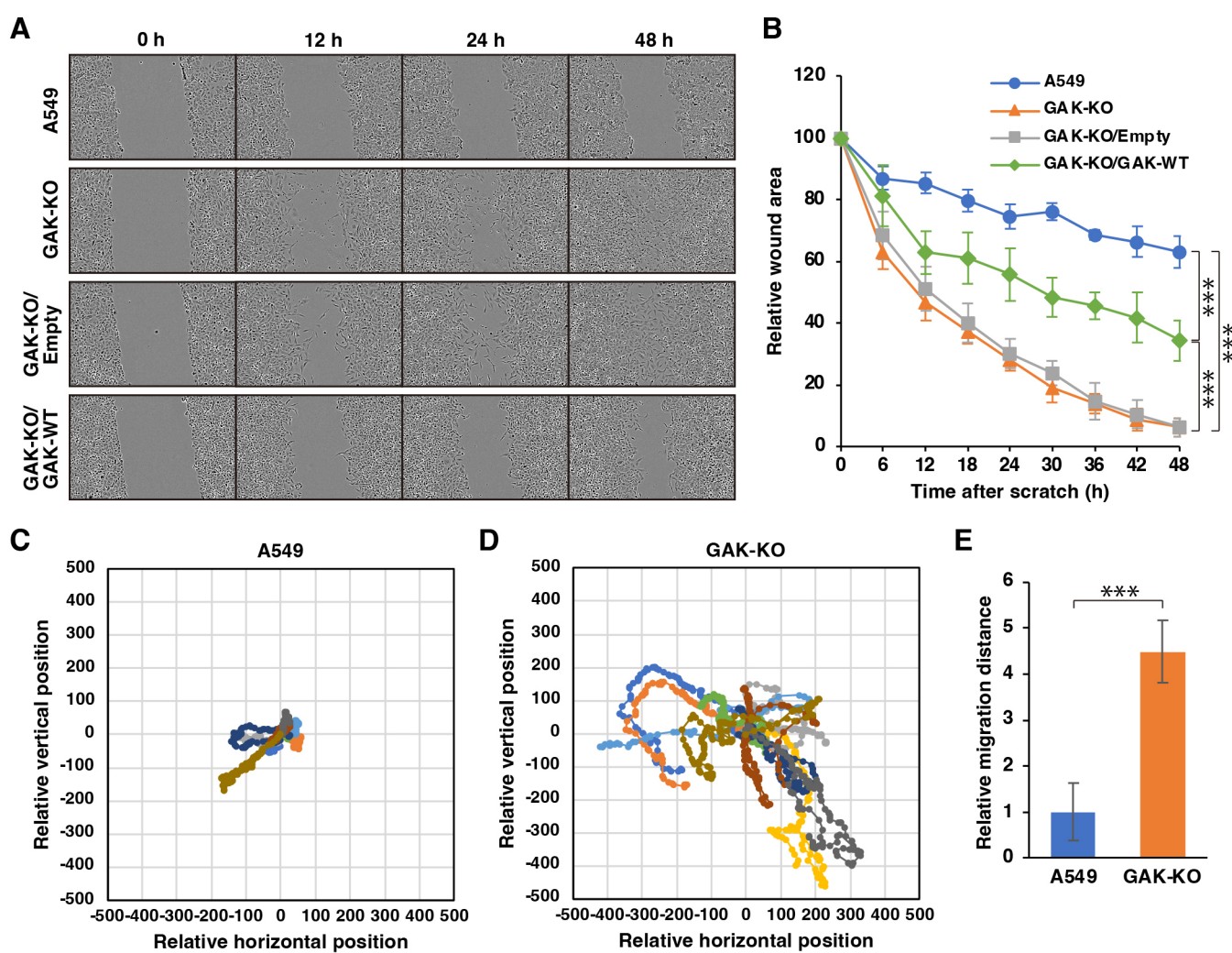

**Fig. 2. GAK disruption promotes cell migration.** (A) Representative phase-contrast images from a wound-healing assay of A549 WT, GAK-KO, GAK-KO/ Empty, and GAK-KO/GAK-WT cells at 0, 12, 24 and 48 h after scratching. (B) Quantification of wound area after scratching over 48 h. Data are presented as mean±s.d. (*n*=6). ***P<0.001 (one-way ANOVA followed by Tukey–Kramer post hoc test). (C,D) Rose plots showing migration tracks of A549 WT (C) and GAK-KO (D) cells. Each line represents the track of an individual cell over 24 h. (E) Quantification of migration distance after 24 h. Data are presented as mean±s.d. (*n*=10). ***P<0.001 (unpaired two-tailed Student's *t*-test).

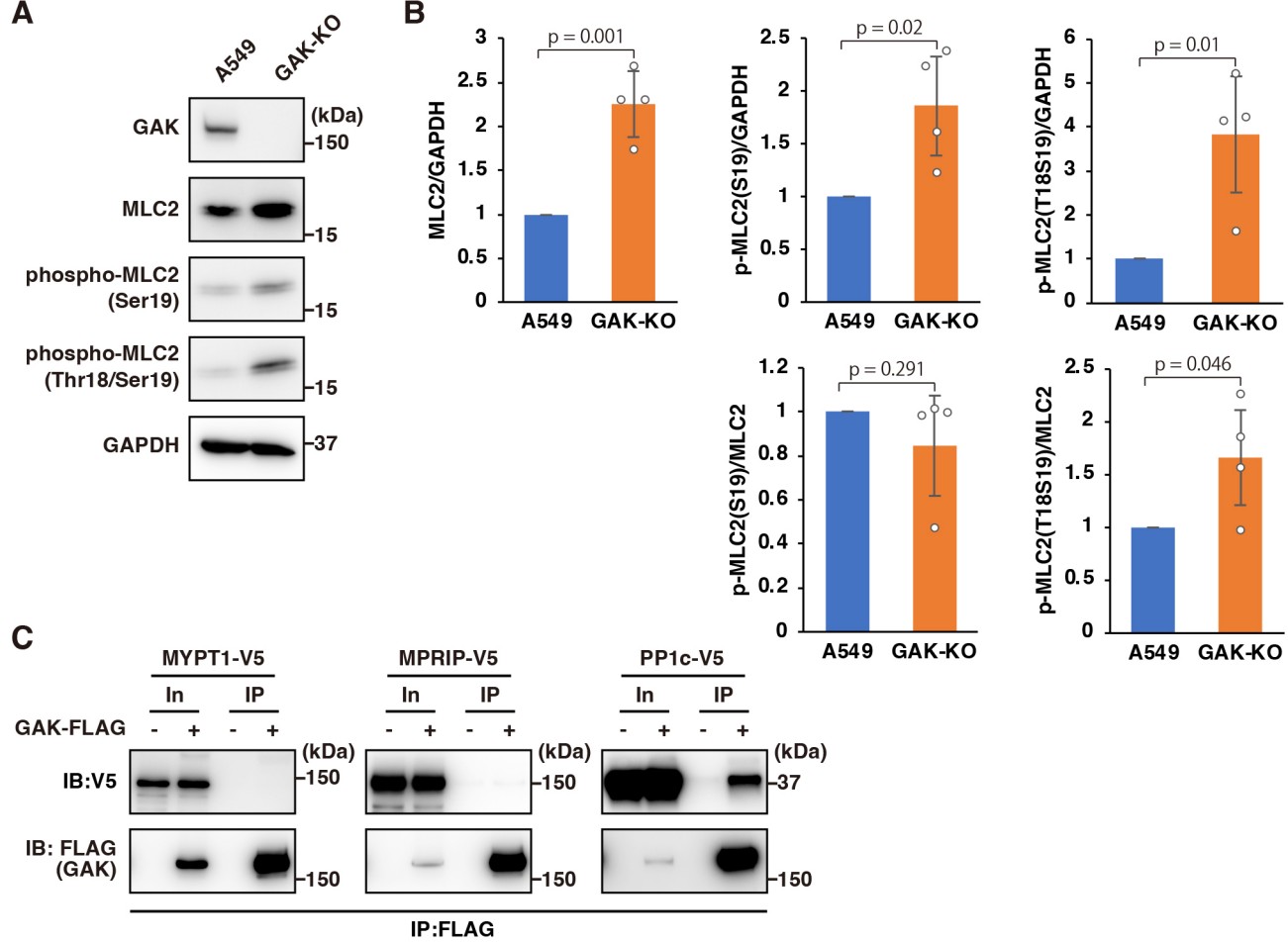

**Fig. 3. GAK disruption promotes MLC phosphorylation.** (A) IB analysis of MLC (denoted MLC2 in the figure) and phosphorylated MLC2 in A549 WT and GAK-KO cells. Representative blots are shown. (B) Quantification of the relative band intensity of total MLC2, phosphorylated MLC2 (Ser19) and phosphorylated MLC2 (Thr18/Ser19), normalised to GAPDH or total MLC2. Data are presented as mean±s.d. (*n*=4; unpaired two-tailed Student's *t*-test). (C) Immunoprecipitation (IP) using an anti-FLAG antibody was performed on lysates from 293T cells overexpressing FLAG-tagged GAK and V5-tagged MYPT1, MPRIP or the catalytic subunit of protein phosphatase 1 (PP1c), followed by immunoblotting (IB) with anti-V5 and anti-FLAG antibodies. In, Input (10%). Representative blots from at least three independent experimental repeats are shown.

phosphorylation coincided with an increase in total MLC expression, thus indicating that Ser19 phosphorylation was not enhanced (Fig. 3B). In contrast, the increase in Thr18 phosphorylation was greater than that in total MLC expression, indicating enhanced Thr18 phosphorylation (Fig. 3B; Fig. S2B). Furthermore, treatment with the GAK-specific inhibitor SGC-GAK-1 enhanced the phosphorylation of Thr18 and Ser19 in MLC (Fig. S2C,D). Because the dual phosphorylation of Thr18 and Ser19 is considered an indicator of stress fibre stabilisation and increased cellular contractility (Vicente-Manzanares et al., 2009; Watanabe et al., 2007), these data support the thicker stress fibres and increased cell motility observed in GAK-KO cells. Cofilin proteins, which depolymerise actin filaments, are inactivated by phosphorylation at Ser3, preventing binding to F-actin and thereby stabilising stress fibres (Arber et al., 1998; Yang et al., 1998). Consistent with this, phosphorylation of cofilin at Ser3 was also elevated in GAK-KO cells (Fig. S2E). Given that GAK has been reported to interact with the regulatory subunit of protein phosphatase 2A, B′γ (Naito et al., 2012; Sato et al., 2009), we next investigated its interaction with each subunit of the myosin phosphatase complex, which dephosphorylates MLC (Somlyo and Somlyo, 2003). We did not detect any interaction between GAK and the regulatory subunits of myosin phosphatase, MYPT1 or MPRIP, but observed an interaction

between GAK and the catalytic subunit, PP1c (PPP1CA) (Fig. 3C). These findings suggest that the interaction between GAK and PP1c may play a role in the regulation of actomyosin by GAK.

### Crosstalk with ROCK-dependent actomyosin regulation

Our previous study has demonstrated that the morphological changes in autophagosomes and autolysosomes induced by GAK-KO were attenuated by ROCK inhibitors or ROCK1 knockdown (Miyazaki et al., 2021). ROCK is also involved in the regulation of actomyosin structures, including stress fibre formation, by directly phosphorylating MLC and MYPT1 (Julian and Olson, 2014; Narumiya et al., 2009; Narumiya and Thumkeo, 2018; O'Connor and Chen, 2013). Therefore, we examined whether stress fibre formation induced by GAK-KO could be attenuated by ROCK inhibition or knockdown. Stress fibre formation in GAK-KO cells was suppressed by the ROCK inhibitor Y-27632 in a dose-dependent manner and also by fasudil, a clinically used ROCK inhibitor for treating cerebral vasospasm in individuals with subarachnoid haemorrhage (Fig. 4A) (Nagumo et al., 2000; Nakamura et al., 2001; Satoh et al., 2012). Additionally, stress fibre formation in GAK-KO cells was suppressed by knockdown of either ROCK1 or ROCK2 (Fig. 4B,C). Furthermore, treatment with the ROCK

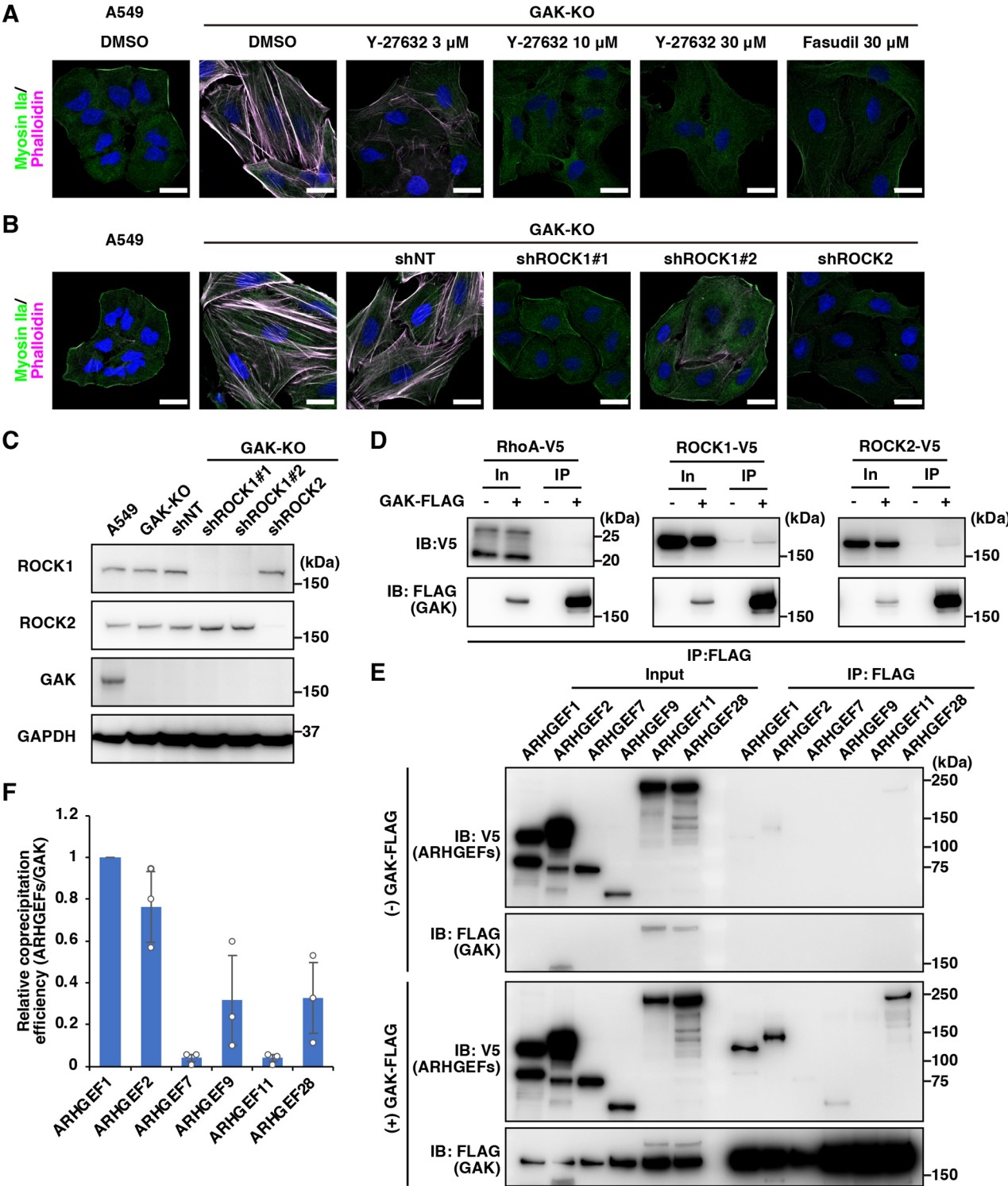

**Fig. 4. Crosstalk with ROCK-dependent actomyosin regulation.** (A) IF microscopy analysis of A549 WT and GAK-KO cells stained for myosin IIa (green in merge), F-actin (phalloidin; magenta in merge), and DNA (DAPI; blue). GAK-KO cells were treated with DMSO, Y-27632 (3–30 µM), or fasudil (30 µM) for 24 h. Scale bars: 20 µm. (B) IF microscopy analysis of A549 WT and GAK-KO cells stained for myosin IIa (green in merge), F-actin (phalloidin; magenta in merge), and DNA (DAPI; blue). GAK-KO cells were transduced with non-targeting shRNA (GAK-KO/shNT), ROCK1-targeted shRNA (GAK-KO/shROCK1#1 and GAK-KO/shROCK1#2), or ROCK2-targeted shRNA (GAK-KO/shROCK2). Scale bars: 20 µm. For A and B, representative images from at least three independent experimental repeats are shown. (C) IB analysis of ROCK1 and ROCK2 expression in A549 WT, GAK-KO, GAK-KO/shNT, GAK-KO/shROCK1#1, GAK-KO/shROCK1#2, and GAK-KO/shROCK2 cells. (D) Immunoprecipitation (IP) with anti-FLAG antibody was performed on lysates from 293T cells overexpressing FLAG-tagged GAK and V5-tagged RhoA, ROCK1 or ROCK2, followed by IB analysis with anti-V5 and anti-FLAG antibodies. For C and D, representative blots from at least three independent experimental repeats are shown. (E) Immunoprecipitation with anti-FLAG antibody was performed on lysates from 293T cells overexpressing FLAG-tagged GAK and V5-tagged Rho guanine nucleotide exchange factors (RhoGEFs) – ARHGEF1, ARHGEF2, ARHGEF7, ARHGEF9, ARHGEF11, or ARHGEF28 – followed by IB analysis with anti-V5 and anti-FLAG antibodies. (F) Quantification of the relative co-precipitation efficiency of ARHGEFs with GAK, with the co-precipitation efficiency of ARHGEF1 with GAK set to 1. Data are presented as mean±s.d. (n=3). In, Input (10%).

inhibitors (Y-27632 or fasudil) completely suppressed increased MLC phosphorylation in GAK-KO cells (Fig. S3A,B). Additionally, knockdown of ROCK1 or ROCK2 attenuated the increased phosphorylation of MLC in GAK-KO cells (Fig. S3C,D). These results suggest that ROCK signalling is activated in GAK-KO cells. We first examined whether GAK interacts with RhoA, ROCK1 or ROCK2 but observed no interaction with either of them (Fig. 4D). We then tested for interactions between GAK and RhoGEFs involved in RhoA activation and identified interactions with several RhoGEFs, including ARHGEF1 and ARHGEF2 (Fig. 4E,F). These

results suggest that GAK might regulate actomyosin dynamics via interactions with RhoGEFs.

### GAK phosphorylates the myosin phosphatase regulatory subunit MYPT1

GAK exhibits kinase activity, and its known substrate consensus motif – threonine residues followed by glycine on the carboxyl side, such as Thr156 in the AP2M1 protein – has been previously reported (Lin et al., 2018). This consensus sequence is also present in ARHGEF2, MPRIP and MYPT1 (Fig. 5A).

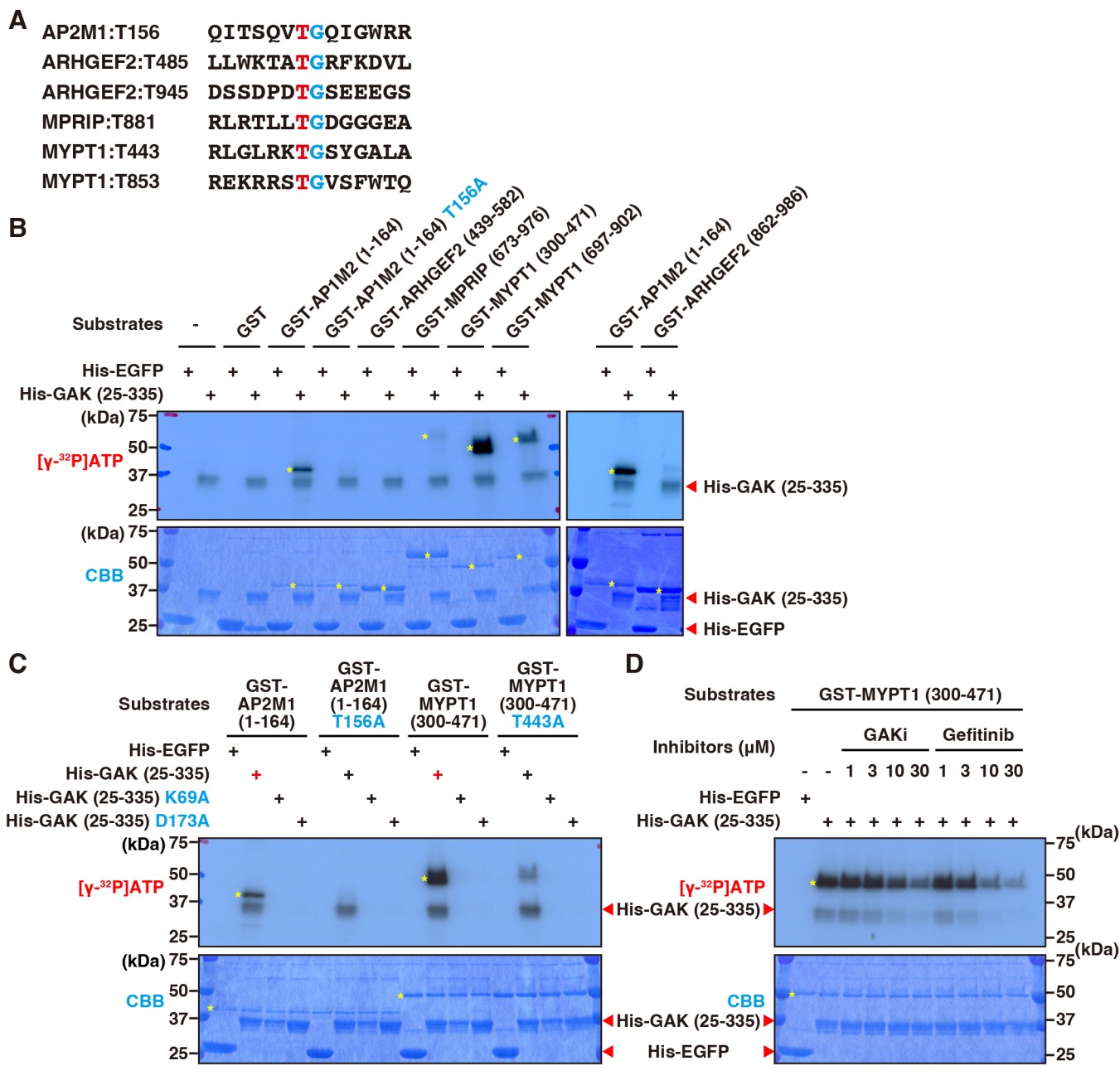

Fig. 5. GAK phosphorylates MYPT1. (A) Amino acid sequences surrounding threonine residues selected as potential phosphorylation sites by GAK. (B) Representative autoradiography and Coomassie Brilliant Blue (CBB) staining following in vitro kinase assays using GAK (amino acid residues 25–335) as the enzyme and ARHGEF2, MPRIP or MYPT1 as potential substrates. (C) Representative autoradiography and CBB staining following in vitro kinase assays using WT GAK or its kinase-dead mutants (K69A and D173A) (amino acid residues 25–335) as the enzyme, and WT AP2M1 or its threonine-to-alanine mutant (amino acid residues 1–164) or WT MYPT1 or its threonine-to-alanine mutant (amino acid residues 300–471) as the substrate. (D) Representative autoradiography and CBB staining following in vitro kinase assays using GAK (amino acid residues 25–335) as the enzyme and MYPT1 (amino acid residues 300–471) as the substrate in the presence of GAK inhibitor (1–30 µM) or gefitinib (1–30 µM). Yellow asterisks highlight the bands of interest in each lane. Representative gels from at least three independent experimental repeats are shown.

Notably, phosphorylation at Thr945 in ARHGEF2 and Thr443 in MYPT1 has been reported, although the kinase responsible for this phosphorylation remains unidentified (Mayya et al., 2009; Zhou et al., 2013). *In vitro* kinase assays revealed that GAK efficiently phosphorylates the region of MYPT1 containing Thr443 (amino acid residues 300–471) (Fig. 5B). Although GAK also phosphorylated the region of MYPT1 containing Thr853 (residues 697–902) and of MPRIP containing Thr881 (residues 673–976), the phosphorylation efficiency was substantially lower than that of the Thr443-containing region of MYPT1 (Fig. 5B). Mutations in the ATP-binding site (K67A) or catalytic site (D173A) of GAK abolished its kinase activity toward the Thr443-containing region of MYPT1 (Fig. 5C). Furthermore, replacing Thr443 with an alanine in this region significantly reduced MYPT1 phosphorylation by GAK (Fig. 5C). Additionally, treatment with a GAK inhibitor or with gefitinib suppressed GAK-induced phosphorylation of this region of MYPT1 in a concentration-dependent manner (Fig. 5D). These results indicate that GAK phosphorylates Thr443 of MYPT1 *in vitro*, suggesting a potential role for this phosphorylation in actomyosin regulation.

### The catalytic activity of GAK plays a limited role in actomyosin regulation

In addition to its kinase domain, GAK contains a phosphatase and tensin homolog (PTEN)-like domain, an intrinsically disordered region (IDR) known to bind clathrin, and a J domain that interacts with heat-shock proteins (Park et al., 2015). To clarify the contribution of the GAK kinase domain to actomyosin regulation, we generated a point mutant within the kinase domain (D173A) and a series of deletion mutants: delM1 (a kinase domain deletion), delM2 (a PTEN-like domain deletion) and delM3 (an IDR deletion) (Fig. 6A). However, no expression was detected in the J domain deletion mutant. We reintroduced either GAK-WT or these mutants into GAK-KO cells and assessed stress fibre formation using phalloidin staining (Fig. 6B). Stress fibres were almost completely eliminated upon reintroduction of GAK-WT, were partially reduced with D173A, delM1 or delM2, and only minimally affected with delM3 (Fig. 6C). To further examine the morphological changes associated with stress fibre formation, we quantified the cell spread area and aspect ratio (Elosegui-Artola et al., 2014). GAK-KO cells exhibited increased spread area and aspect ratio, whereas the reintroduction of GAK-WT reduced both metrics to levels comparable to those in parental cells (Fig. 6D,E). Furthermore, similar to the phalloidin fluorescence intensity results, D173A, delM1 and delM2 partially reduced the spread area and aspect ratio, whereas delM3 had only a minimal effect (Fig. 6D,E). In order to analyse the contribution of GAK kinase activity to actomyosin regulation at different angles, we performed analyses using MYPT1 mutants at Thr443, a residue that has been shown to be phosphorylated by GAK, including a phosphorylation-deficient form (TA) and phosphorylation-mimicking forms (TD or TE). When wild-type or mutant MYPT1 was transduced into A549 cells, no significant changes were observed in MLC phosphorylation or stress fibre formation (Fig. S4). Next, when MYPT1 was transduced into the GAK-KO cells, no significant change in MLC phosphorylation was observed, but significant attenuation of stress fibre formation was observed only when the wild-type or TE mutants were introduced (Fig. S5). Collectively, these findings suggest that the kinase activity of GAK contributes distinctly, but only partially, to actomyosin regulation, whereas the IDR of GAK plays a more prominent role.

### The IDR of GAK regulates actomyosin dynamics through interaction with ARHGEF2

Next, we investigated the role of each GAK domain in regulating MLC phosphorylation. Phosphorylation of MLC, which was elevated in GAK-KO cells, was suppressed to almost the same level as that in the parental cells upon reintroduction of GAK-WT, but not by reintroducing the IDR mutant delM3 (Fig. 7A,B). To further examine each domain of GAK, we analysed binding between the catalytic subunit of myosin phosphatase and the GAK deletion mutants. Co-precipitation of PP1c (PPP1CA) was greater with delM1 than with GAK-WT, whereas no difference was observed between WT and delM3 (Fig. 7C,D). In contrast, co-precipitation of ARHGEF2 was significantly decreased with delM3 compared to GAK-WT (Fig. 7E, F). For ARHGEF1, no difference was observed between GAK-WT and delM3 (Fig. S6A,B). Furthermore, GAK and ARHGEF2 have been reported to localise to the Golgi (Callow et al., 2005; Greener et al., 2000), and in this study, colocalisation of GAK and ARHGEF2 with TGOLN2, a Golgi marker protein, was observed (Fig. S6C). Analysis of wild-type and mutant GAK reintroduced into GAK-KO cells revealed that GAK-WT and delM1 colocalised with ARHGEF2 in the Golgi (Fig. S6D). However, in delM2, colocalisation with ARHGEF2 was observed, but neither GAK nor ARHGEF2 was found to be localised in the Golgi (Fig. S6D). Furthermore, in delM3, ARHGEF2 was localised to the Golgi, whereas GAK was not, and colocalisation of GAK and ARHGEF2 was barely observed (Fig. S6D). Taken together, these results suggest that the reduced ability of delM3 to suppress MLC phosphorylation and stress fibre formation is due to weakened interaction with ARHGEF2. Previous studies have reported that phosphorylation of Ser886 in ARHGEF2 promotes its binding to 14-3-3 proteins, sequestering ARHGEF2 in an inactive state on microtubules (Birkenfeld et al., 2007; Meiri et al., 2014; Zenke et al., 2004). Consistent with this, we found that phosphorylation of Ser886 in ARHGEF2 was decreased in GAK-KO cells (Fig. 7G,H), indicating increased guanine nucleotide exchange activity of ARHGEF2. Consequently, knockdown of ARHGEF2 in GAK-KO cells led to suppressed stress fibre formation and reduced spread area and aspect ratio (Fig. 7I–L; Fig. S6E). Furthermore, knockdown of ARHGEF2 significantly suppressed the increased phosphorylation of MLC in GAK-KO cells (Fig. S6F,G). In addition, ARHGEF1 knockdown significantly suppressed the increased phosphorylation of MLC and attenuated stress fibre formation in GAK-KO cells; however, there was no significant difference in the cell spread area or aspect ratio (Fig. S6H–N). Although the ARHGEF1-binding domain in GAK and changes in ARHGEF1 activity in GAK-KO cells remain to be elucidated, these results suggest that ARHGEF1 is also involved in the regulation of actomyosin by GAK.

Next, we analysed intracellular Rho activity by assessing the amount of activated RhoA bound to the RhoA-binding domain of rhotekin, an effector protein of RhoA (Ren et al., 1999). The assay system was confirmed to work because increased levels of activated RhoA were detected in the lysates of A549 and GAK-KO cells upon treatment with GTPγS; however, no significant difference in the amount of activated RhoA was detected between the A549 and GAK-KO cells (Fig. S7A,B). Collectively, these findings suggest that although RhoA activation cannot be confirmed in GAK-KO cells, GAK inhibits ARHGEF2 activity through interaction via its IDR, thereby regulating actomyosin dynamics.

### The IDR of GAK regulates actomyosin dynamics via the regulation of MLC gene expression

In GAK-KO cells, not only was MLC phosphorylation increased, but also the expression level of MLC was increased (Fig. 3A,B).

Journal of Cell Science

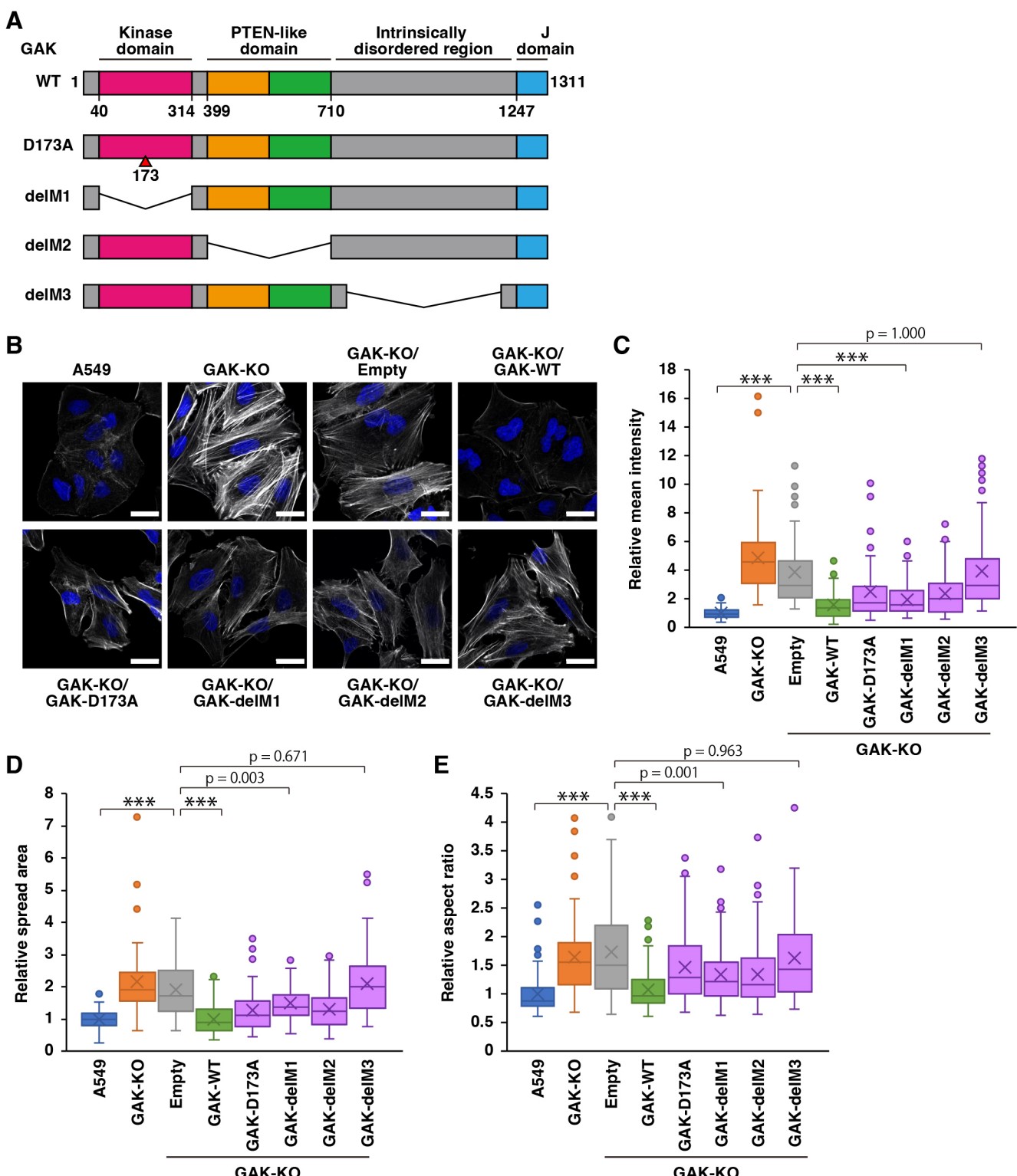

**Fig. 6. Catalytic activity of GAK plays a limited role in actomyosin regulation.** (A) Schematic of GAK and its mutants generated in this study. (B) F-actin (phalloidin) staining in A549 WT, GAK-KO, and GAK-KO cells transduced with an empty vector (Empty), GAK-WT, or GAK mutants (GAK-D173A, GAK-delM1, GAK-delM2, or GAK-delM3). Scale bars: 20 µm. (C–E) Quantification of mean phalloidin intensity (C), cell spread area (D) and aspect ratio (E). At least 60 cells per condition were quantified from three independent experiments and are presented in box plots. The box extends from the lower to the upper quartile; the middle line indicates the median; the cross indicates the mean; and the whiskers represent the minimum to maximum values, except for outliers. Data points outside 1.5× interquartile range from the quartiles are considered outliers and are shown as dots. ***$P<0.001$ (one-way ANOVA followed by Tukey–Kramer post hoc test).

Therefore, we analysed the relationship between GAK and regulation of MLC expression. To verify whether this effect occurred at the transcriptional level, we measured the expression level of *MYL9* mRNA, which encodes the major MLC in non-muscle cells (Park et al., 2011). The results showed that increased *MYL9* mRNA expression was observed in all the three GAK-KO cell lines (Fig. 8A).

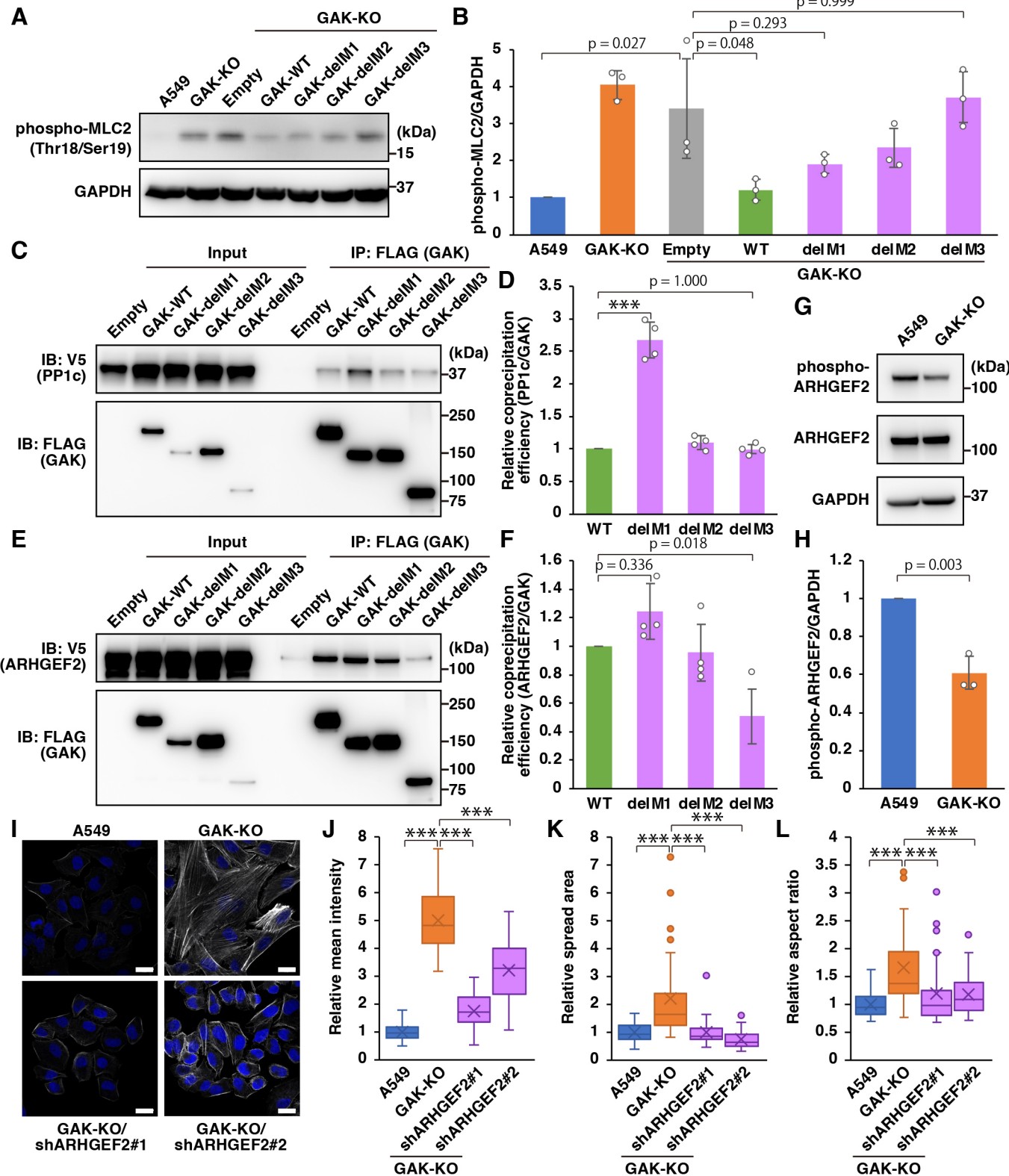

**Fig. 7.** See next page for legend.

Additionally, although no significant changes were detected in the mRNA expression of other regulatory and essential light chains, a significant increase was detected in the mRNA expression of the myosin heavy chain *MYH9* (Fig. S7C–E). Furthermore, increased *MYL9* mRNA expression in GAK-KO cells was significantly

suppressed by the reintroduction of GAK-WT. Significant suppression was also observed with the reintroduction of delM1 and delM2; however, the reintroduction of the IDR mutant delM3 resulted in little suppression (Fig. 8B). Finally, the knockdown of *MYL9* in GAK-KO cells substantially suppressed stress fibre

**Fig. 7. The IDR of GAK is involved in actomyosin regulation through interaction with ARHGEF2.** (A) IB analysis of phosphorylated MLC (denoted MLC2 in the figure) (Thr18/Ser19) in A549 WT, GAK-KO and GAK-KO cells transduced with an empty vector (Empty), GAK-WT or GAK mutants (delM1, delM2 or delM3). (B) Quantification of the relative band intensity of phosphorylated MLC2 (Thr18/Ser19), normalised to GAPDH. Data are presented as mean±s.d. (*n*=3; one-way ANOVA followed by Tukey–Kramer post hoc test). (C) Immunoprecipitation (IP) with anti-FLAG antibody was performed on lysates from 293T cells overexpressing FLAG-tagged GAK (WT or mutants) and V5-tagged PP1c, followed by IB analysis with anti-V5 and anti-FLAG antibodies. (D) Quantification of the relative co-precipitation efficiency of PP1c with GAK (WT or mutants). Data are presented as mean±s.d. (*n*=4). \*\*\**P*<0.001 (one-way ANOVA followed by Tukey–Kramer post hoc test). (E) Immunoprecipitation with anti-FLAG antibody was performed on lysates from 293T cells overexpressing FLAG-tagged GAK (WT or mutants) and V5-tagged ARHGEF2, followed by IB analysis with anti-V5 and anti-FLAG antibodies. Inputs are 10% in C and E. (F) Quantification of the relative co-precipitation efficiency of ARHGEF2 with GAK (WT or mutants). Data are presented as mean±s.d. (*n*=4; one-way ANOVA followed by Tukey–Kramer post hoc test). (G) IB analysis of ARHGEF2 and phosphorylated ARHGEF2 (Ser886) in A549 WT and GAK-KO cells. (H) Quantification of the relative band intensity of phosphorylated ARHGEF2, normalised to GAPDH. Data are presented as the mean±s.d. (*n*=3; unpaired two-tailed Student's *t*-test). (I) F-actin (phalloidin) staining in A549 WT, GAK-KO and GAK-KO cells transduced with ARHGEF2-targeted shRNA (GAK-KO/shARHGEF2#1 and GAK-KO/shARHGEF2#2). Scale bars: 20 μm. (J–L) Quantification of phalloidin mean intensity (J), cell spread area (K) and cell aspect ratio (L). At least 30 cells per condition were quantified from three independent experiments and are presented in box plots. The box extends from the lower to the upper quartile; the middle line indicates the median; the cross indicates the mean; and the whiskers represent the minimum to maximum values, except for outliers. Data points outside 1.5× interquartile range from the quartiles are considered outliers and are shown as dots. \*\*\**P*<0.001 (one-way ANOVA followed by Tukey–Kramer post hoc test).

formation, although no significant changes were observed in the cell spread area or the aspect ratio (Fig. 8C,D; Fig. S7F–H). Taken together, these findings suggest that GAK IDR also plays an important role in regulating *MYL9* expression and that it is one of the crucial components of the mechanism through which GAK controls actomyosin dynamics.

## DISCUSSION
Actin filaments are involved in numerous cellular processes, and abnormalities in actin dynamics are associated with pathological disorders, such as cancer and neurodegenerative diseases (Bamburg and Wiggan, 2002; Cingolani and Goda, 2008; Yamaguchi and Condeelis, 2007). Therefore, understanding actin cytoskeleton dynamics is essential not only for elucidating the underlying mechanisms of several physiological and pathological processes but also for developing therapeutic strategies for related diseases. In this study, we found that knocking out GAK in cultured cells leads to prominent stress fibre formation and a significant increase in cell motility (Fig. S8).

Yeast kinases Ark1p and Prk1p, which share amino acid sequence conservation in their kinase domains with GAK, have been reported to regulate actin organisation and endocytosis (Henry et al., 2003; Sekiya-Kawasaki et al., 2003; Toshima et al., 2005). Although GAK has been implicated in clathrin-mediated endocytosis, to the best of our knowledge, there have been no reports of its role in actin regulation. In our previous study, we showed that GAK and ROCK signalling exhibit antagonistic effects on autophagic flux regulation (Miyazaki et al., 2021). In this study, we demonstrated that GAK also antagonises the ROCK-dependent regulation of actomyosin dynamics, specifically stress fibre formation. Furthermore, it has been suggested that the antagonistic effects of GAK are mediated through

multiple pathways. One pathway involves suppression of ARHGEF2 activity through the interaction of GAK with ARHGEF2. Although phosphorylation of Ser886 – associated with ARHGEF2 inhibition – was suppressed in GAK-KO cells, this site in ARHGEF2 does not correspond to a known GAK substrate consensus sequence, suggesting that GAK is unlikely to directly phosphorylate Ser886 in ARHGEF2. Further investigation is needed to clarify how GAK suppresses ARHGEF2 activity. Another pathway involves GAK regulation of total MLC expression; however, the mechanism underlying this regulation remains to be elucidated.

The findings of this study are as follows. First, GAK-KO cells exhibited increased MLC phosphorylation and stress fibre formation, which led to enhanced cell motility. These phenotypes were eliminated by the reintroduction of GAK. Among GAK domains, the intrinsically disordered region has been identified as a major contributor to actomyosin regulation. Second, the phenotype of GAK-KO cells was consistently attenuated by the pharmacological inhibition or knockdown of ROCK, indicating that GAK and ROCK function antagonistically to regulate actomyosin dynamics. The increased phosphorylation of cofilin at Ser3 further supports enhanced ROCK pathway activity in GAK-KO cells. Thirdly, ARHGEF2 contributed to the GAK-KO phenotype, as ARHGEF2 knockdown significantly diminished actomyosin-related changes. Reduced phosphorylation of ARHGEF2 at Ser886 was consistent with the increased ARHGEF2 activity in GAK-KO cells. Fourthly, despite these findings, biochemical analysis using the RBD pulldown assay did not detect an increase in global RhoA activity in GAK-KO cells. Although we cannot exclude the possibility that RhoA activation occurs in a highly transient or spatially restricted manner, which is not captured by bulk biochemical assays, our data do not support sustained whole-cell activation of RhoA as the primary mechanism underlying the GAK-KO phenotype. Notably, several RhoA-independent mechanisms for ROCK activation have been reported, including lipid-mediated activation via the PH domain and Rho-independent conformational activation of ROCK (Araki et al., 2001; Truebestein et al., 2015). Additionally, ARHGEF2 has been reported to regulate actomyosin via the activation of RhoA, as well as other small G proteins such as Rac1 and Cdc42 (Ren et al., 1998; Waheed et al., 2013). Furthermore, ARHGEF2 functions independently of GEF activity and is involved in RAS-MAPK signalling, focal adhesion formation and anisotropic stress fibre orientation (Cullis et al., 2014; Huang et al., 2014). These observations provide plausible explanations for how ROCK and ARHGEF2 might contribute independently or cooperatively to actomyosin regulation in GAK-KO cells, even in the absence of detectable RhoA activation.

The Thr443 residue of MYPT1 is phosphorylated by GAK *in vitro*; however, the physiological significance of this phosphorylation is not completely understood. Introduction of a phosphorylation-mimicking MYPT1 TE mutant suppressed stress fibre formation in GAK-KO cells, suggesting that phosphorylation of Thr443 of MYPT1 might lead to MYPT1 activation. Additionally, the expression level of the MYPT1 TE mutant after transduction was suppressed compared to that of the wild-type and other mutants, suggesting that phosphorylation of Thr443 might induce a conformational change in MYPT1, leading to increased activity and accelerated degradation. Given that methylation of Lys442 in MYPT1 has been reported to contribute to MYPT1 stabilisation (Cho et al., 2011), it is worth investigating how Thr443 phosphorylation affects Lys442 methylation and vice versa, thereby contributing to MYPT1 stability. Furthermore, the correlation between MLC phosphorylation, stress fibre formation, cell spread area and aspect

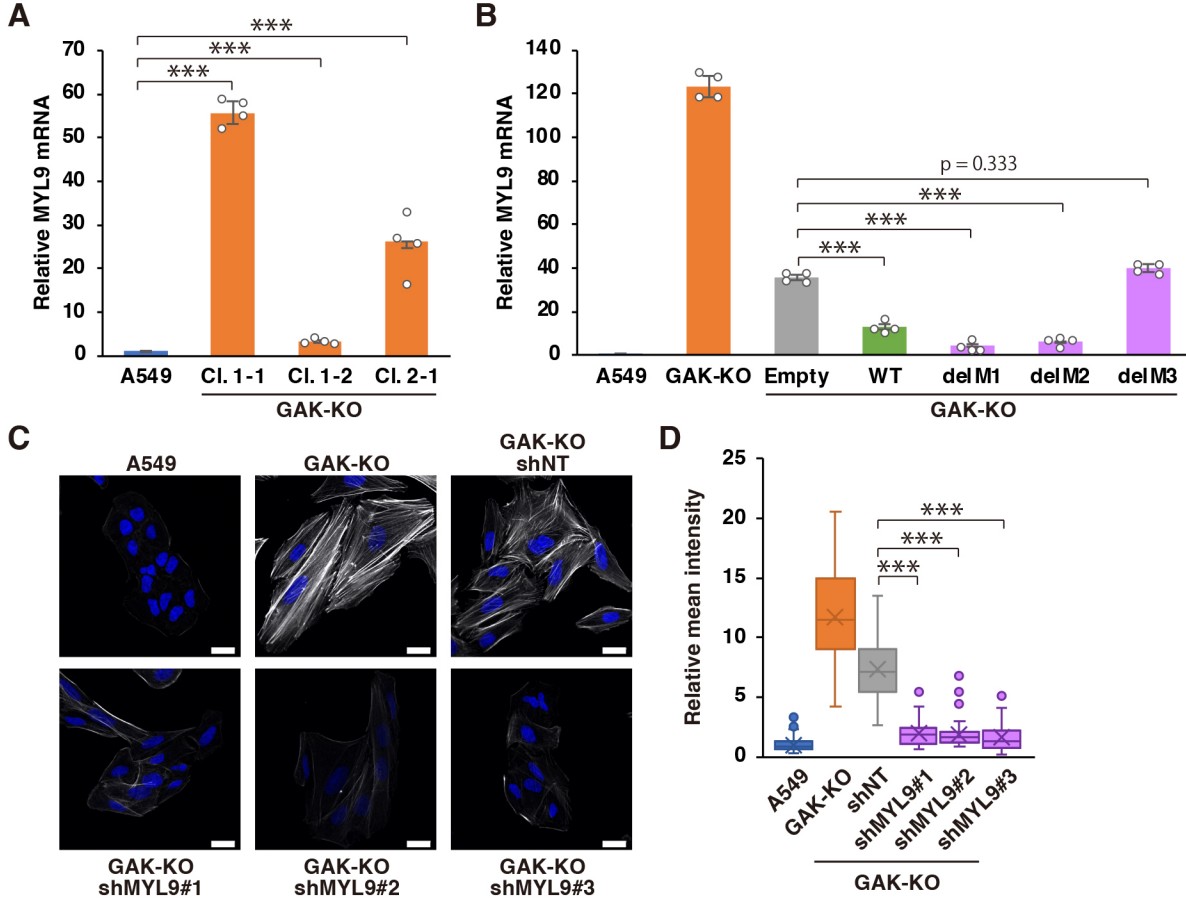

**Fig. 8. The IDR of GAK regulates actomyosin dynamics via the regulation of MLC gene expression.** (A) RT-qPCR analysis of the relative mRNA expression levels of *MYL9* in A549 WT and GAK-KO cells (clones 1-1, 1-2, and 2-1). *GAPDH* was used as an internal control. Data are presented as mean±s.d. (*n*=4). ***$P<0.001$ (one-way ANOVA followed by Tukey–Kramer post hoc test). (B) RT-qPCR analysis of relative mRNA expression levels of *MYL9* in A549 WT, GAK-KO and GAK-KO cells transduced with an empty vector (Empty), GAK-WT, or GAK mutants (delM1, delM2 or delM3). *GAPDH* was used as an internal control. Data are presented as mean±s.d. (*n*=4). ***$P<0.001$ (one-way ANOVA followed by Tukey–Kramer post hoc test). (C) F-actin (phalloidin) staining in A549 WT, GAK-KO and GAK-KO cells transduced with non-targeting shRNA (A549/shNT) or MYL9-targeted shRNA (GAK-KO/ shMYL9#1, GAK-KO/shMYL9#2, and GAK-KO/shMYL9#3). Scale bars: 20 μm. (D) Quantification of phalloidin mean intensity. At least 30 cells per condition were quantified from three independent experiments and are presented in box plots. The box extends from the lower to the upper quartile; the middle line indicates the median; the cross indicates the mean; and the whiskers represent the minimum to maximum values, except for outliers. Data points outside 1.5× interquartile range from the quartiles are considered outliers and are shown as dots. ***$P<0.001$ (one-way ANOVA followed by Tukey–Kramer post hoc test).

ratio was not clear upon the introduction of MYPT1 (Figs S4 and S5), or ROCK knockdown (Fig. 3B; Fig. S3C,D) in GAK-KO cells. Therefore, the functional relationship between MYPT1, ROCK and GAK in actomyosin regulation requires further investigation.

SNP rs1564282, located in intron 1 of the *GAK* gene, has been reported to increase the risk of Parkinson's disease (UK Parkinson's Disease Consortium et al., 2011; Dumitriu et al., 2011; Ma et al., 2015; Pankratz et al., 2009; Rhodes et al., 2011). Moreover, several *GAK* SNPs appear to act synergistically in the development of sporadic Parkinson's disease through multiple pathways (Zhang et al., 2016). Besides, mouse models of Parkinson's disease have shown that microglial phagocytosis of degenerated dopaminergic neurones is mediated by ROCK-Cdc42 signalling and that ROCK inhibition prevents phagocytic neuronal elimination (Barcia et al., 2012). In contrast, mice with microglia-specific RhoA deletion exhibit microglial activation and develop neurological phenotypes, including synaptic and neuronal loss (Socodato et al., 2020). Furthermore, the RhoA-Src signalling pathway has been found to be suppressed in microglia in mouse models of Alzheimer's disease (Socodato et al., 2020). Collectively, these findings suggest that fine-tuning of Rho-ROCK signalling is essential for maintaining

neuronal homeostasis. Therefore, further studies are warranted to elucidate the role of GAK in microglial function in relation to the ROCK pathway and its potential involvement in the pathogenesis of neurodegenerative diseases such as Parkinson's disease.

Cell migration is essential for physiological processes, such as development and wound healing, but also plays a crucial role in pathological conditions, including cancer metastasis (Trepat et al., 2012). Recent studies have shown that FBXO22 inhibits the proliferation and metastasis of cervical cancer cells through the ubiquitin-dependent degradation of GAK (Li et al., 2023). At first glance, this finding seems inconsistent with our results, where GAK KO in the A549 lung cancer cell line led to increased cell motility. However, under the complex *in vivo* environment, cancer cells employ diverse migration modes, including mesenchymal, amoeboid and collective migration, each regulated by actin cytoskeletal reorganisation, actomyosin contractility and Rho-ROCK signalling (Friedl and Wolf, 2010; Te Boekhorst et al., 2016). Therefore, further investigation is required to clarify the role of GAK in these distinct modes of migration. Moreover, GAK expression has been reported to be upregulated in osteosarcoma cells and hormone-refractory prostate cancer cells (Ray et al., 2006;

Sakurai et al., 2014; Susa et al., 2010). In contrast, data from Gene Expression Profiling Interactive Analysis 2 (GEPIA2) has revealed that GAK expression is elevated in tumour tissues compared to in normal tissues in certain cancers, including pancreatic adenocarcinoma and cholangiocarcinoma, whereas it is reduced in others, such as lung adenocarcinoma and lung squamous cell carcinoma. Therefore, the role of GAK in cancer cell metastasis, particularly across different tumour types, needs to be investigated *in vivo*, particularly using mouse and zebrafish models.

Another factor must be considered in relation to cancer. Tyrosine kinase inhibitors, such as gefitinib and dasatinib, are effectively used as molecular-targeted drugs for cancer; however, serious adverse events, such as interstitial lung disease characterized by inflammation and fibrosis, can rarely occur (Ando et al., 2006; Camus et al., 2004; Inoue et al., 2003; Ito et al., 2019; Kim et al., 2022; Weatherald et al., 2020). In this study, mild stress fibre formation was observed even with gefitinib, which inhibits the kinase activity of GAK, although to a lesser extent than that observed in GAK knockout or knockdown cells. If tyrosine kinase inhibitors, such as gefitinib have a similar effect on lung interstitial cells, it is possible that inhibition of GAK kinase activity as an off-target effect may contribute to the onset of interstitial lung disease; therefore, careful investigation is warranted.

In this study, we demonstrated that GAK antagonises the ROCK-dependent regulation of actomyosin dynamics by suppressing ARHGEF2 activity and MLC expression, thereby regulating actomyosin contractility and cell migration. Our study provides novel insights into the molecular mechanisms controlling cytoskeletal organisation and cell motility. Understanding the role of GAK in these processes may have broader implications for physiological processes such as development and wound healing, as well as pathological conditions including neurodegenerative diseases and cancer metastasis.

## MATERIALS AND METHODS
### Reagents
Y-27632 (cat. no. 688000), hydroxyfasudil (cat. no. 390602) and GAK inhibitor (cat. no. 538770) were purchased from Calbiochem (Darmstadt, Germany), and DAPI and guanosine-5′-(γ-thio)triphosphate tetralithium salt (GTPγS) were obtained from Sigma-Aldrich (St. Louis, MO, USA). Gefitinib (cat. no. 13166) was purchased from Cayman Chemical (Ann Arbor, MI, USA), and SGC-GAK-1 (cat. no. 6564) was obtained from Tocris Biosciences (Bristol, UK). Protease inhibitor cocktail, phosphatase inhibitor cocktail and isopropyl-β-D-thiogalactopyranoside (IPTG) were obtained from Nacalai Tesque (Kyoto, Japan). Blasticidin S (cat. no. 029-18701), G418 (cat. no. 078-05961), and puromycin (cat. no. 166-23153) were purchased from Wako Pure Chemical (Osaka, Japan). EzStain AQua (cat. no. AE-1340; ATTO, Tokyo, Japan) was used to stain SDS-polyacrylamide gels. Acti-stain 555 phalloidin (cat. no. PHDH1; Cytoskeleton, Denver, CO, USA) was used for fluorescent staining of F-actin. Oligonucleotides used in this study are given in Tables S1–S6.

### Antibodies
The following primary antibodies were used in this study: anti-GAK [cat. no. M057-3; lot no. 022; 1:500 for immunoblotting (IB), 1:100 for immunofluorescence (IF)] from Medical & Biological Laboratories (Aichi, Japan); anti-phospho-MLC (MLC2; Ser19) (cat. no. 3671; lot no. 6; 1:1000 for IB), anti-phospho-MLC (MLC2; Thr18/Ser19) (cat. no. 3674; lot no. 5; 1:1000 for IB), anti-MLC (MLC2; cat. no. 8505; lot no. 4; 1:1000 for IB), anti-phospho-cofilin (Ser3) (cat. no. 3311; lot no. 11; 1:1000 for IB), anti-cofilin (cat. no. 5175; lot no. 4; 1:1000 for IB), anti-phospho-ARHGEF2 (Ser886) (cat. no. 14143; lot no. 1; 1:1000 for IB), anti-ARHGEF2 (cat. no. 4076; lot no. 3; 1:1000 for IB), anti-ARHGEF1 (cat. no. 3669; lot no. 3; 1:1000 for IB), anti-RhoA (cat. no. 2117; lot no. 5; 1:1000 for IB), anti-ROCK1

(cat. no. 4035; lot no. 2; 1:1000 for IB), anti-ROCK2 (cat. no. 9029; lot no. 3; 1:1000 for IB), anti-myosin IIa (cat. no. 3403; lot no. 2; 1:50 for IF), and anti-α-tubulin (cat. no. 3873; lot no. 15; 1:1000 for IF) from Cell Signaling Technology (Danvers, MA, USA); anti-GAPDH (cat. no. sc-32233; lot no. K0510; 1:2000 for IB) and anti-vimentin (cat. no. sc-6260; lot no. L1312; 1:200 for IF) from Santa Cruz Biotechnology (Dallas, TX, USA); anti-cytokeratin 18 (cat. no. 628402; lot no. B174672; 1:125 for IF) from BioLegend (San Diego, CA, USA); anti-ARHGEF2 (cat. no. ab155785; lot no. GR253664-33; 1:200 for IF) from Abcam (Cambridge, UK); anti-TGOLN2/TGN46 (cat. no. ab155785; lot no. 150743; 1:500 for IF) from Bio-Rad Laboratories (Hercules, CA, USA); anti-FLAG (cat. no. F1804; 1:1000 for IB, 1:500 for IF) from Sigma-Aldrich, and anti-V5 (R960-25; 1:5000 for IB) from Invitrogen (Carlsbad, CA, USA). Horseradish peroxidase (HRP)-conjugated secondary antibodies against mouse IgG (cat. no. 115-035-003; 1:3000 for IB) and rabbit IgG (cat. no. 711-035-152; 1:3000 for IB) were purchased from Jackson ImmunoResearch (West Grove, PA, USA). Alexa Fluor 488-conjugated secondary antibody against rabbit IgG (cat. no. A11034; 1:1000 for IF), Alexa Fluor 488-conjugated secondary antibody against sheep IgG (cat. no. A11015; 1:1000 for IF), Alexa Fluor 555-conjugated secondary antibody against mouse IgG (cat. no. A21424; 1:1000 for IF), Alexa Fluor 647-conjugated secondary antibody against rabbit IgG (cat. no. A21244; 1:1000 for IF) were obtained from Invitrogen (Eugene, OR, USA).

### Cell culture
The human lung cancer cell line A549 (ATCC No. CCL-185), the human pancreatic cancer cell line PANC-1 (ATCC No. CRL-1469) and the human liver cancer cell line HepG2 (ATCC No. HB-8065) were purchased from American Type Culture Collection (ATCC; Manassas, VA, USA) and cultured in RPMI 1640 medium (Merck Sigma-Aldrich) supplemented with 10% heat-inactivated foetal bovine serum (FBS; Gibco, Grand Island, NY, USA) and 1% penicillin-streptomycin solution (Wako Pure Chemical) at 37°C in a humidified atmosphere with 5% $CO_2$. The human oral squamous cell carcinoma line CAL27 (ATCC No. CRL-2095) and human embryonic kidney cell line 293T (ATCC No. CRL-3216) were also obtained from ATCC and cultured in Dulbecco's modified Eagle's medium (DMEM) containing high glucose (Sigma-Aldrich) and supplemented with 10% heat-inactivated FBS and 1% penicillin-streptomycin solution at 37°C in a humidified atmosphere with 5% $CO_2$. The human neuroblastoma cell line SH-SY5Y (cat. no. 94030304) was obtained from the European Collection of Authenticated Cell Cultures and cultured in DMEM containing low glucose (Sigma-Aldrich) and supplemented with 10% heat-inactivated FBS and 1% penicillin-streptomycin solution at 37°C in a humidified atmosphere with 5% $CO_2$. All cell lines were authenticated by short tandem repeat DNA genotype analysis at BEX (Tokyo, Japan) and routinely tested for mycoplasma using the e-Myco Mycoplasma PCR-based detection kit (iNtRON Biotechnology, Inc., Gyeonggi, Korea).

### Construction of plasmids
The cDNA for human *GAK* was cloned into the mammalian expression vector pcDNA6/V5-His (Invitrogen, Carlsbad, CA, USA) as described in our previous study (Miyazaki et al., 2021). Deletion or point mutants of *GAK* were generated via site-directed mutagenesis using a PCR-based method with PrimeSTAR HS DNA polymerase (Takara Bio, Shiga, Japan), and the sequences were verified by Sanger sequencing at FASMAC (Kanagawa, Japan). DNA fragment encoding GAK was then subcloned into the pLentiN lentiviral expression vector (Addgene #37444, deposited by Karl Munger; Spangle et al., 2012).

To construct the lentiviral short hairpin RNA (shRNA) vectors, oligonucleotides for shRNAs targeting genes of interest (*GAK*, *ROCK1*, *ROCK2*, *ARHGEF2* and *MYL9*) at appropriate sites and non-targeting controls were synthesised (Table S1). These oligonucleotides were phosphorylated, annealed, and inserted into the pLKO.1 puro lentiviral shRNA vector (Addgene #8453, deposited by Bob Weinberg; Stewart et al., 2003).

To construct the mammalian expression vectors, cDNA encoding the genes of interest (*MYPT1*, *MPRIP*, *PPP1CA*, *RHOA*, *ROCK1*, *ROCK2*, *ARHGEF1*, *ARHGEF2*, *ARHGEF7*, *ARHGEF9*, *ARHGEF11* and *ARHGEF28*) was amplified via PCR using the PrimeSTAR HS DNA polymerase and cloned into pcDNA6/V5-His. The cloned sequences were verified by Sanger

Journal of Cell Science

sequencing at FASMAC. Point mutants of *MYPT1* were generated via site-directed mutagenesis using a PCR-based method with PrimeSTAR HS DNA polymerase, and the sequences were verified by Sanger sequencing at FASMAC. The DNA fragment encoding MYPT1 was then subcloned into the pLentiN lentiviral expression vector.

To produce recombinant proteins of the GAK kinase domain (residues 25–335) and EGFP, DNA fragments encoding these proteins were amplified via PCR using the PrimeSTAR HS DNA polymerase and cloned into pCold II (Takara Bio Inc., Shiga, Japan). Finally, to generate the GST-tagged proteins for the candidate regions phosphorylated by GAK, DNA fragments encoding these regions (AP2M1, residues 1–164; ARHGEF2, residues 439–582 and 862–986; MPRIP, residues 673–976; MYPT1, residues 300–471 and 697–902) were amplified via PCR using the PrimeSTAR HS DNA polymerase and cloned into pGEX-4T1 (GE Healthcare, Uppsala, Sweden).

### Generation of stable cell lines

To establish cell lines expressing fluorescent proteins, a DNA fragment encoding tdTomato was subcloned from the tdTomato-C1 vector (Addgene #54653, deposited by Michael Davidson), into the pcDNA6/V5-His vector. A549 cells were transfected with the plasmid pcDNA6-tdTomato using Lipofectamine 3000 (Invitrogen) according to the manufacturer's instructions. Additionally, A549/GAK-KO cells were transfected with the plasmid pEGFP-N1 (Clontech, Mountain View, CA, USA) using Lipofectamine 3000. After selecting the transfected A549 cells using blasticidin S (10 µg/ml) or A549/GAK-KO cells with G418 (500 µg/ml), single clones were isolated, and the expression of tdTomato or EGFP, respectively, was confirmed using a fluorescent microscope. After expansion of the clones, these A549/tdTomato or A549/GAK-KO/EGFP cells were cultured under the same conditions as the A549 parental cells, except that blasticidin S (5 µg/ml) or G418 (250 µg/ml), respectively, was maintained in the culture medium for selection.

To generate stable cell lines expressing WT or mutant *GAK*, the production and transduction of lentiviral particles were carried out as previously described (Miyazaki et al., 2021). Briefly, 293T cells were co-transfected with the constructed plasmid (pLentiN-GAK-WT or mutants), pMD2.G (Addgene #12259; deposited by Didier Trono) and psPAX2 (Addgene #12260, deposited by Didier Trono) using PEI-MAX (Polysciences, Warrington, PA, USA) and cultured at 37°C in a humidified atmosphere with 5% $CO_2$. The culture supernatant containing the lentiviral particles was harvested 48 h post-transfection. A549/GAK-KO cells were transduced overnight with the lentivirus-containing supernatant and selected using blasticidin S (5 µg/ml). The established cell lines were cultured under the same conditions as the A549 parental cells, except that blasticidin S (2.5 µg/ml) was maintained in the culture medium for selection. In order to generate stable cell lines expressing WT or mutant MYPT1, lentiviral particle production and transduction were performed as previously described for GAK. A549 or A549/GAK-KO cells were transduced overnight with the lentivirus-containing supernatant and selected using blasticidin S (10 µg/ml for A549 and 5 µg/ml for A549/GAK-KO). The established cell lines were cultured under the same conditions as the A549 parental cells, except that blasticidin S (5 µg/ml for A549 and 2.5 µg/ml for A549/GAK-KO) was maintained in the culture medium for selection.

To establish cell lines that suppress the expression of a gene of interest, lentiviral particles were generated as described above, except that the pLKO.1 puro vector was used instead of the pLentiN vector. Cells were transduced overnight with the prepared lentiviruses and selected using puromycin (2 µg/ml for A549 cells or 1 µg/ml for A549/GAK-KO cells). The established cell lines were cultured under the same conditions as the A549 cells, except for the addition of puromycin (1 µg/ml for A549 cells or 0.5 µg/ml for A549/GAK-KO cells) to maintain selection pressure.

### IF analysis

Cells were seeded onto 13 mm diameter glass coverslips placed in a 24-well culture plate and incubated for 48 h. After washing with phosphate-buffered saline (PBS), cells were fixed with 2% paraformaldehyde in PBS for 10 min at room temperature (RT, 20–25°C) and permeabilised by incubation with 0.1% Triton X–100 for 5 min at RT. After washing with Tris-buffered saline

containing 0.05% Tween 20 (TBS-T), cells were blocked with 10% FBS in TBS-T at RT for 1 h, followed by incubation with the primary antibody in TBS-T supplemented with 1.5% FBS and 0.1% bovine serum albumin at 4°C overnight (12–16 h). After washing with TBS-T, cells were incubated with the Alexa Fluor–conjugated secondary antibodies, DAPI (1 µM) and phalloidin (100 nM) at 37°C for 1 h. After a final wash with TBS-T, the coverslips were mounted using ProLong Diamond Glass Mountant (Invitrogen, Eugene, OR, USA). The cells were visualised using a Zeiss LSM700 confocal laser scanning fluorescence microscope (Zeiss GmbH, Jena, Germany) equipped with Plan-Apochromat 63×/1.4 Oil DIC or 40×/1.4 Oil DIC objectives (Zeiss GmbH). All images were acquired and processed identically using ZEN 2012 software (version 8.1.0.484; Zeiss GmbH). Object-based fluorescence intensity was quantified using ImageJ software (version 1.50i; National Institutes of Health). In this study, whole-cell integrated fluorescence was used as the primary metric without dedicated segmentation or background subtraction procedures, because the phalloidin fluorescence in untreated cells was nearly indistinguishable from the background.

### IB analysis

Total cellular proteins were extracted using a lysis buffer containing 50 mM Tris-HCl pH 8.0, 150 mM NaCl, 1.0% Nonidet P-40 (NP-40), 0.5% sodium deoxycholate, 0.1% SDS, 1% protease inhibitor cocktail and 1% phosphatase inhibitor cocktail. Samples were sonicated to disrupt protein aggregates using a Branson 450D Sonifier (Emerson, Danbury, CT, USA). Protein concentrations were measured using the BCA Protein Assay Kit (Thermo Fisher Scientific, Rockford, IL, USA) according to the manufacturer's instructions. Equal amounts of protein were resolved by SDS-polyacrylamide gel electrophoresis (SDS-PAGE) and transferred onto Immobilon-P membranes (Merck Millipore, Cork, Ireland). The membranes were blocked with 5% non-fat milk and probed with primary antibodies. Immunoreactive proteins were detected using HRP-conjugated secondary antibodies and the Immobilon Western HRP substrate detection reagent (Merck Millipore, Billerica, MA, USA). Densitometric analysis was performed using a WSE-6300 Luminograph III molecular imager (ATTO Corporation, Tokyo, Japan). Uncropped images of blots from this paper are shown in Fig. S9.

### Immunoprecipitation

Mammalian expression vectors were transfected into 293T cells using PEI-MAX. At 48 h post-transfection, cells were lysed in lysis buffer (50 mM Tris-HCl pH 8.0, 150 mM NaCl and 1.0% NP-40) supplemented with 1% protease inhibitor cocktail. Samples were sonicated as described in the previous section. Immunoprecipitation from cell lysates was performed using anti-FLAG M2 magnetic beads (cat. no. M8823; Sigma-Aldrich), and the beads were washed three times with lysis buffer. Co-precipitated proteins were eluted from the beads with lysis buffer supplemented with 100 µg/ml of 3×FLAG peptide. Immunoprecipitates or total cell lysates were analysed via IB with an anti-FLAG antibody (1:1000) and anti-V5 antibody (1:5000), as described in the above section.

### Transmission electron microscopy

Cells were initially fixed with 2.5% glutaraldehyde in 0.1 M phosphate buffer (pH 7.3) for 1 h at 4°C. They were subsequently fixed in 1% osmium tetroxide for 1 h at 4°C, dehydrated through a graded ethanol series (30–100%), and embedded in Quetol 812 epoxy resin (Nisshin EM, Tokyo, Japan) for 2 days at 60°C. Ultrathin sections (60 nm) were prepared using an Ultracut J microtome (Reichert Jung, Vienna, Austria). The sections were stained with uranyl acetate (Merck, Darmstadt, Germany) for 15 min at RT in a dark box. After rinsing with water, the sections were further stained with lead nitrate for 10 min at RT and imaged using a transmission electron microscope (JEM-1200EX II; JEOL, Tokyo, Japan). All images were captured on electron microscopy film (FG type; Fujifilm).

### Wound-healing assay

Cells were seeded in six-well plates and cultured under standard conditions (10% FBS) until confluent. A wound was created in the cell monolayer using a sterile 200 µl pipette tip. The detached cells were washed and

incubated in a reduced-serum medium (1% FBS) to minimise the effects of proliferation. Images of the wound area were captured at 0 h and every hour for up to 48 h using the IncuCyte ZOOM cell imaging system (Sartorius Essen BioScience, Ann Arbor, MI, USA). Quantification of the wound area was performed using ImageJ.

### Random migration assay

Cells were seeded at low density in glass-bottom culture dishes (Greiner Bio-One, Frickenhausen, Germany) and allowed to adhere overnight. The cells were imaged every 15 min for 24 h using the Zeiss LSM700 confocal laser-scanning fluorescence microscope equipped with a Plan-Apochromat 10×/0.45 objective. Cell movement was manually tracked, and migration distances were measured using ImageJ (Manual Tracking plugin).

### Preparation of recombinant proteins

*Escherichia coli* BL21 (DE3) cells (Thermo Fisher Scientific) transformed with pCold II-GAK (25–335) or pCold II-EGFP were cultured at 37°C until reaching an $OD_{600}$ 0.4–0.6. Cultures were then transferred to 15°C and further incubated for 4 h after the addition of 0.5 mM IPTG. The cells were pelleted by centrifugation (5000 *g* for 15 min), resuspended in His-tag lysis buffer (20 mM Tris-HCl pH 8.0, 500 mM NaCl and 0.1% NP-40), and lysed by sonication using the Branson 450D Sonifier. The lysate was centrifuged to remove the insoluble fraction, and the resulting supernatant was used to purify His-tagged proteins using TALON metal affinity resin (Clontech) according to the manufacturer's instructions.

The same *E. coli* BL21 (DE3) cells were also transformed with pGEX-4T1 series constructs and incubated at 37°C until reaching $OD_{600}$ 0.4–0.6. Cultures were then transferred to 20°C and incubated for an additional 3 h after the addition of 0.5 mM IPTG. The cells were pelleted by centrifugation (5000 *g* for 15 min), resuspended in a GST-tag lysis buffer (50 mM Tris-HCl pH 8.0, 500 mM NaCl, 1 mM EDTA and 0.5% NP-40), and lysed by sonication using the Branson 450D Sonifier. The insoluble fraction was removed by centrifugation (20,000 *g* for 10 min), and the supernatant was used to purify GST-tagged proteins using Glutathione Sepharose 4B resin (GE Healthcare, Uppsala, Sweden), according to the manufacturer's instructions.

The purified recombinant proteins were dialysed against a dialysis buffer (20 mM HEPES-NaOH pH 7.9, 10% glycerol, 100 mM KCl, 0.2 mM EDTA, 1 mM $MgCl_2$, 0.2 mM $CaCl_2$, 0.1% NP-40, 1 mM DTT and 0.5 mM PMSF), aliquoted and stored at −80°C until use.

### *In vitro* kinase assay

GST-tagged proteins were incubated in a kinase reaction buffer (20 mM HEPES-NaOH pH 7.9, 100 mM KCl, 10 mM $MgCl_2$, 5 mM MnCl2, 1 mM DTT and 20 µM ATP) with His-tagged GAK (residues 25–335) or EGFP and 25 µCi [γ-$^{32}$P]ATP (cat. no. NEG502H; PerkinElmer, Waltham, MA, USA) at 30°C for 30 min. The reaction was stopped by adding 2× SDS sample buffer and boiling at 98°C for 5 min. Phosphorylated proteins were separated by SDS-PAGE. The gels were stained with Coomassie Brilliant Blue (EzStain AQua; ATTO), dried, and subjected to autoradiography using X-ray film (SUPER HR-HA; Fujifilm).

### Rho activity analysis

To produce a GST fusion protein containing the RhoA-binding domain [amino acids 7–89] of Rhotekin (GST-RBD), the DNA fragment encoding this domain was amplified by PCR using PrimeSTAR HS DNA polymerase and cloned into pGEX-4T1 (GE Healthcare, Uppsala, Sweden). Expression of GST fusion proteins in *E. coli* BL21 (DE3) and purification of GST fusion proteins using Glutathione Sepharose 4B resin were performed as previously described. Active RhoA pull-down experiments were performed as described by Ren et al. (1999), with slight modifications. Briefly, A549 and A549/GAK-KO cells were lysed in buffer A (50 mM Tris-HCl pH 7.5, 1% Triton X-100, 0.5% sodium deoxycholate, 0.1% SDS, 500 mM NaCl, 20 mM $MgCl_2$ and 1% protease inhibitor cocktail). The cell lysates were clarified by centrifugation at 14,000 *g* at 4°C for 10 min, and equal volumes of lysates were incubated with GST-RBD (50 µg) beads at 4°C for 60 min. The beads were washed four times with buffer B (25 mM Tris-HCl [pH 7.5] containing 40 mM NaCl, 30 mM $MgCl_2$ and 1% protease inhibitor cocktail). For the positive control samples, GTPγS was added to a final concentration of 0.1 mM and then incubated at 30°C for 15 min. The bound Rho proteins were detected by immunoblotting using a monoclonal antibody against RhoA. The amount of RBD-bound Rho was normalised to the total amount of Rho in cell lysates to compare Rho activity (level of GTP-bound Rho) in different samples.

### Quantitative RT-PCR

Total RNA was extracted from A549 and A549/GAK-KO cells using the NucleoSpin RNA Plus Kit (Takara Bio, Shiga, Japan) and reverse transcribed into cDNA using the PrimeScript RT Master Mix Kit (Takara Bio) according to the manufacturer's instructions. Target gene expression was determined by quantitative PCR (qPCR) using the TB Green Premix Ex Taq II (Tli RNaseH Plus) Kit (Takara Bio). The sequences of validated primers are listed in Table S6. qPCR was performed in a CFX Opus 96 system (Bio-Rad Laboratories) under the following conditions: initial cDNA denaturation at 95°C for 30 s, followed by 40 cycles of denaturation at 95°C for 5 s and simultaneous annealing and extension at 60°C for 30 s. The obtained data were analysed using the Bio-Rad CFX Maestro 2.3 Software (Bio-Rad Laboratories), and the comparative *Ct* method ($2^{-\Delta\Delta Ct}$) was used for relative quantification of gene expression (Livak and Schmittgen, 2001). Data were standardised using *GAPDH* as an internal control.

### Statistical analysis

All data are presented as mean±s.d. from at least three independent experiments. Statistical analyses were conducted using the SPSS version 29 software (IBM, Armonk, NY, USA), and significance was determined using an unpaired two-tailed Student's *t*-test or one-way analysis of variance followed by a Tukey–Kramer post hoc test. $P<0.05$ was considered statistically significant.

### Acknowledgements

The authors thank Dr Shota Moriya, Ms. Ayako Hirota, Ms. Mayumi Tokuhisa, and Ms. Jun Takemura (Department of Biochemistry, Tokyo Medical University, Tokyo, Japan) for their technical assistance. We are grateful to the Electron Microscopy Core Facility at our institution for their comprehensive assistance, from sample preparation to the acquisition of electron microscopy images. We would like to thank Editage (www.editage.jp) for English language editing.

### Competing interests

The authors declare no competing or financial interests.

### Author contributions

Conceptualization: M.H.; Formal analysis: M.H., N.T., H. Kazama, H. Hino; Funding acquisition: M.H., K.M.; Investigation: M.H., N.T., H. Kazama, H. Hino; Resources: H. Kokuba; Supervision: H. Handa, K.M.; Visualization: M.H., H. Kazama; Writing – original draft: M.H.; Writing – review & editing: M.H., N.T., H. Handa, K.M.

### Funding

This work was supported by the Japan Society for the Promotion of Science KAKENHI (grant numbers 18K06901, 21K07106 and 25K11442 to M.H.), the MEXT-Supported Program of the Strategic Research Foundation at Private Universities (grant number S1411011 to K.M.) from the Ministry of Education, Culture, Sports, Science and Technology of Japan. Open Access funding provided by Tokyo Medical University. Deposited in PMC for immediate release.

### Data and resource availability

Any material generated in this study, such as plasmids, can be obtained from the corresponding author upon reasonable request. All relevant data and details of resources can be found within the article and its supplementary information.

### Peer review history

The peer review history is available online at https://journals.biologists.com/jcs/lookup/doi/10.1242/jcs.264117.reviewer-comments.pdf

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
