## [Peer Review File · Journal of Cell Science]

GAK antagonises ROCK-dependent regulation of actomyosin dynamics

Naoharu Takano, Hiroko Kokuba, Hiromi Kazama, Hirotsugu Hino, Hiroshi Handa, Keisuke Miyazawa and Masaki Hiramoto
DOI: 10.1242/jcs.264117

Editor: Michael Way

Review timeline

Original submission:	30 April 2025
Editorial decision:	10 June 2025
First revision received:	16 February 2026
Accepted:	2 March 2026

Original submission

First decision letter

MS ID#: jcs.264117

MS TITLE: GAK antagonises Rho-ROCK signalling to regulate actomyosin dynamics by suppressing ARHGEF2

AUTHORS: Masaki Hiramoto; Naoharu Takano; Hiroko Kokuba; Hiromi Kazama; Hirotsugu Hino; Hiroshi Handa; Keisuke Miyazawa

ARTICLE TYPE: Research Article

Dear Dr Hiramoto,

We have now reached a decision on the above manuscript.

To see the reviewers' reports and a copy of this decision letter, please go to:

As you will see, the two reviewers are positive but raise a number of criticisms / questions that prevent me from accepting the paper at this stage. They suggest, however, that a revised version might prove acceptable, if you can address their concerns. In particular, based on their comments, I think it is important to perform RhoA pull-downs or MYPT1 phosphorylation (as a ROCK readout) for a readout of RhoA activity as well as address reviewer 2 question concerning the point about Gefitinib. If you think that you can deal satisfactorily with the criticisms on revision, I would be pleased to see a revised manuscript.

Reviewer 1

SUMMARY OF THE ADVANCE MADE IN THIS PAPER AND ITS POTENTIAL SIGNIFICANCE TO THE FIELD

This study describes the role of GAK in controlling actomyosin dynamics in A549 cells. The main findings are: GAK depletion increases actomyosin fiber formation and decreases motility, by increasing MLC phosphorylation in a ROCK-dependent manner. Data suggests that MYPT1 phosphorylation by GAK could be involved, although the data there is not as definitive. The authors also chop GAK into pieces and discover that the kinase domain is essentially dispensable for the effect of GAK on cytoskeletal dynamics, but interaction with ARHGEF2 is not.

SUGGESTIONS TO AUTHORS

Overall, this is an impressive, very solid paper that nonetheless requires some essential revisions. If the authors provide the following, I'll be happy to recommend its acceptance.

- * When the authors refer to GAK-KO cells, are these clonal cells? Mixed populations? Mixtures of clones? This needs to be stated in the results.
- * Figure 3 (Western blots) needs to be quantified. Also, GAK KO cells seem to contain more MLC, not only phosphorylated MLC. Is this reproducible? Or clone-dependent?
- * What is the effect of MYPT1 T443A or T443D on MLC phosphorylation and stress fiber formation?
- * Since the kinase-less (and others) mutant of GAK has a similar effect on cytoskeletal dynamics than the wild type form and the authors seem to indicate that this is due to the interaction of GAK with ARHGEF2, the authors need to address the localization and co-localization of these molecules in the A549 cells by indirect immunofluorescence or similar. FRET is not required at this stage, although it could be interesting for a future manuscript.

Reviewer 2

SUMMARY OF THE ADVANCE MADE IN THIS PAPER AND ITS POTENTIAL SIGNIFICANCE TO THE FIELD

Building on prior research showing G-associated kinase (GAK) inhibits Rho-ROCK signaling, Miyazawa and colleagues investigated the molecular mechanisms behind this in their study. They used stress fiber formation as a readout for Rho/ROCK activity and observed that GAK's kinase domain isn't a primary contributor to this inhibitory effect. Instead, they showed that the IDR domain of GAK interacts with and inhibits the RhoA GEF Arhgef2, leading to RhoA/ROCK inhibition. While the significance might appear limited given that GAK was already known to inhibit the RhoA/ROCK pathway, the authors' findings reveal a new mechanism of RhoA inhibition, which could be of interest to the readership. The experiments are well-described, and most of the conclusions appear sound. However, some aspects—such as the involvement of specific GAK domains and the rationale for focusing on Arhgef2—are insufficiently addressed. Additional experiments using more specific approaches (beyond stress fiber analysis) are needed to validate some of the findings.

SUGGESTIONS TO AUTHORS

Defining how GAK regulates RhoA/ROCK activity. This section is weakened by two main issues: the difficulty in quantifying stress fibers (SF) due to their heterogeneity, and the lack of direct evidence showing an effect on RhoA/ROCK activity. In Figure delM1, some cells still show substantial stress fibers, and the differences among the domain mutants are not particularly striking in terms of stress fiber phenotype. The authors also discuss effects on MLC phosphorylation and cell area; however, as these parameters are regulated by a wide range of signaling pathways, it could be misleading to use them to solely draw conclusions on RhoA/ROCK pathway activity. Drawing conclusions about GAK's role (and specific GAK domain) in RhoA/ROCK regulation based solely on SF morphology and MLC phosphorylation carries a high risk of misinterpretation. The authors need a more specific readout to consolidate their conclusions, such as FRET RhoA sensor, RBD-pulldown, ROCK kinase activity assays.

In addition, the authors used domain depletion experiments, but it is unclear why kinase inhibitor has not been used (see Fig. 6D,E). Earlier in the manuscript, they report that Gefitinib—a drug known to inhibit GAK as an off-target effect—mimics the impact of GAK knockout on stress fibers (Fig. 1E). However, they later conclude that the kinase domain of GAK is not involved. This apparent contradiction needs to be addressed. Have they tested the effect of a specific GAK inhibitor on stress fibers?

In a similar way, the authors used stress fibers to test the interplay between GAK and Arhgef2, but this approach is highly unspecific. There is no direct testing of Arhgef2 activity or RhoA activity, which is a significant omission given that the main objective of the study is to understand how GAK inhibits RhoA/ROCK.

Given that GAK interacts with additional GEFs (as depicted in Figure 4E), an investigation into the IDR's binding to these other GEFs, beyond just Arhgef2, would be valuable. Although Arhgef2 may

have been the initial focus due to a putative GAK phosphorylation motif, the authors' demonstration that the kinase domain is not a primary contributor necessitates exploring whether IDR-mediated inhibition influences the activity of other GEFs and thereby impacts stress fiber formation. Since there is so much binding and a connection to ROCK, it is tempting to speculate that GAK may directly localize to actomyosin filaments, did the author try to localize GAK subcellularly? Furthermore, no details are provided regarding the quantitative assessment of stress fibers (methods for segmentation, background subtraction, etc.).

First revision

Author response to reviewers' comments

Response to Reviewer 1

We are grateful for the constructive comments that helped us improve our paper. We have considered all comments and suggestions in the revised version of our paper. Our point-by-point responses are as follows.

1. When the authors refer to GAK-KO cells, are these clonal cells? Mixed populations? Mixtures of clones? This needs to be stated in the results.

Response to Comment 1:

Thank you for your comments. As suggested, we have added an explanation in the Results section that when we refer to GAK-KO cells, these are clonal cells, and of the three clonal cells, we usually use clone 1-1. We also changed the order of the original Figure S1A and S1B.

2. Figure 3 (Western blots) needs to be quantified. Also, GAK KO cells seem to contain more MLC, not only phosphorylated MLC. Is this reproducible? Or clone-dependent?

Response to Comment 2:

Thank you for your comments. As suggested, we have quantified the western blot in Figure 3A and presented it in Figure 3B (and renamed the original Fig. 3B Fig. 3C).

The results showed that both MLC phosphorylation and total MLC expression increased. This was also true for all three clones, as shown in the revised Figures S2A and S2B. Furthermore, we have added analyses of increased total MLC expression and presented them in the revised Figure 8. In all three GAK-KO clones, MYL9/MLC2 expression increased at the transcriptional level (Fig. 8A). Furthermore, the reintroduction of wild-type GAK, but not the IDR-deficient delM3 mutant, reduced its expression (Fig. 8B). Moreover, the knockdown of MYL9 in GAK-KO cells attenuated stress fibre formation (Fig. 8C,D). The supplementary figures for this study are presented in Fig. S7C-H.

3. What is the effect of MYPT1 T443A or T443D on MLC phosphorylation and stress fiber formation?

Thank you for your valuable comments. Following your suggestion, we created the MYPT1 phosphorylation-deficient mutant T443A and the phosphorylation- mimicking mutants T443D and T443E and performed additional analyses in which they were transduced into A549 WT or GAK-KO cells. First, transduction into A549 WT cells had little effect on MLC expression, phosphorylation, or stress fibre formation. The results are shown in Figure S4. Transduction into GAK-KO cells had little effect on MLC expression and phosphorylation, but transduction with wild-type or T443E mutant MYPT1 significantly attenuated stress fibre formation in GAK-KO cells. The results are shown in Figure S5. Although no change was observed in the transduction of T443D, transduction with the phosphorylation-mimicking T443E mutant partially attenuated the effect of GAK-KO, further suggesting that GAK-mediated phosphorylation of MYPT1 at T443 plays a role in GAK-mediated actomyosin regulation. In the original manuscript, we have mentioned that the contribution of GAK kinase activity to GAK-mediated actomyosin regulation is partial (although we

have not stated that it does not contribute), but to avoid any misunderstanding that it does not contribute, we have added a statement in the Results section that it does contribute “distinctly” but partially.

4. Since the kinase-less (and others) mutant of GAK has a similar effect on cytoskeletal dynamics than the wild type form and the authors seem to indicate that this is due to the interaction of GAK with ARHGEF2, the authors need to address the localization and co-localization of these molecules in the A549 cells by indirect immunofluorescence or similar. FRET is not required at this stage, although it could be interesting for a future manuscript.

Response to Comment 4:

Thank you for your comments. As suggested, we analysed GAK and ARHGEF2 using indirect immunofluorescence. Because GAK and ARHGEF2 have been reported to localise to the Golgi apparatus, we also used TGOLN2/TGN46 as a Golgi marker protein. In A549 WT cells, GAK and ARHGEF2 were present throughout the cell, but co-localisation in the Golgi apparatus was also observed (Fig. S6C). Next, we analysed the subcellular localisation of GAK mutants reintroduced into GAK-KO cells (Fig. S6D). Wild-type and delM1 mutants showed co-localisation with ARHGEF2 near the nucleus. However, in the delM2 mutant, although GAK and ARHGEF2 co-localised, they were not localised near the nucleus. Furthermore, in the delM3 mutant, ARHGEF2 was localised near the nucleus, whereas GAK was not. No Co-localisation of GAK and ARHGEF2 was observed. These results support the immunoprecipitation results (Figure 7E,F) and suggest that GAK-delM3 is unable to interact with ARHGEF2 and is, therefore, unable to suppress ARHGEF2, resulting in an inability to rescue the phenotype of GAK- KO cells. This supports the importance of the GAK IDR in actomyosin regulation.

Response to Reviewer 2

Thank you for reviewing our manuscript and providing thoughtful comments and suggestions, which have helped us improve it. We have considered all the comments and suggestions in the revised version of our paper. Our point-by-point responses are as follows.

1. Defining how GAK regulates RhoA/ROCK activity. This section is weakened by two main issues: the difficulty in quantifying stress fibers (SF) due to their heterogeneity, and the lack of direct evidence showing an effect on RhoA/ROCK activity. In Figure delM1, some cells still show substantial stress fibers, and the differences among the domain mutants are not particularly striking in terms of stress fiber phenotype. The authors also discuss effects on MLC phosphorylation and cell area; however, as these parameters are regulated by a wide range of signaling pathways, it could be misleading to use them to solely draw conclusions on RhoA/ROCK pathway activity. Drawing conclusions about GAK's role (and specific GAK domain) in RhoA/ROCK regulation based solely on SF morphology and MLC phosphorylation carries a high risk of misinterpretation. The authors need a more specific readout to consolidate their conclusions, such as FRET RhoA sensor, RBD-pulldown, ROCK kinase activity assays.

Response to Comment 1:

Thank you for highlighting the need for more specific evidence to support our conclusions regarding RhoA/ROCK signalling. Accordingly, we directly assessed RhoA activity using an RBD pull-down assay. The assay was validated using GTP γ S-treated controls; however, under our experimental conditions, we did not detect a significant difference in global RhoA activity between A549 WT and GAK-KO cells (Fig. S7A,B).

These data indicated that GAK loss does not measurably alter steady-state RhoA activity at the whole-cell level. We also attempted to evaluate ROCK activity by immunoblotting for MYPT1 Thr696 phosphorylation using two independent antibodies (Cell Signaling Technology, Cat. No. 5163; Medical & Biological Laboratories, Cat. No. CY-M1011). In A549 cells, no specific bands were detected in either the WT or GAK-KO cells, even after calyculin A treatment, thus suggesting that MYPT1 Thr696 phosphorylation is not a reliable indicator of ROCK activity in this cellular context. Importantly, despite the absence of detectable changes in global RhoA activity, multiple independent experiments supported the functional involvement of ROCK signalling in the GAK-KO

phenotype. In particular, cytoskeletal and morphological defects induced by GAK loss were consistently rescued by pharmacological ROCK inhibition and knockdown (Fig. 4A,B, S3A-D). In addition, GAK-KO cells exhibited increased cofilin phosphorylation at Ser3 (Fig. S2E), which was consistent with the enhanced activity of the ROCK-LIMK pathway. Based on these results, we have revised the manuscript to avoid the overinterpretation of stress fibre morphology and MLC phosphorylation as direct indicators of RhoA activity. Instead, we concluded that GAK antagonises the ROCK-dependent regulation of actomyosin dynamics, while any effects on RhoA activity are likely to be transient and/or spatially restricted, and therefore, not detectable by global biochemical assays. In line with this revised interpretation, we have modified the manuscript title to “GAK antagonises ROCK-dependent regulation of actomyosin dynamics”. As the reviewer has noted, resolving the potential localised RhoA dynamics requires approaches such as FRET-based RhoA biosensors, which represent an important direction for future studies.

Additionally, regarding the analysis that previously only evaluated stress fibre formation, we have added an evaluation of MLC phosphorylation (Fig. S2A-D, S3A-D, S6F,G,I,J).

2. In addition, the authors used domain depletion experiments, but it is unclear why kinase inhibitor has not been used (see Fig. 6D,E). Earlier in the manuscript, they report that Gefitinib—a drug known to inhibit GAK as an off-target effect—mimics the impact of GAK knockout on stress fibers (Fig. 1E). However, they later conclude that the kinase domain of GAK is not involved. This apparent contradiction needs to be addressed. Have they tested the effect of a specific GAK inhibitor on stress fibers?

Response to Comment 2:

Thank you for your comments. As suggested, in addition to gefitinib, we examined the effects of the GAK inhibitor used in the *in vitro* kinase assay and another GAK-specific inhibitor, SGC-GAK-1. We observed that, similar to gefitinib, the addition of the GAK inhibitor or SGC-GAK-1 enhanced stress fibre formation (Fig. S1G,H). Furthermore, enhanced MLC phosphorylation was observed, particularly after the addition of SGC-GAK-1 (Fig. S2C,D). Following the suggestion of another reviewer, we performed an analysis using mutants (phosphorylation-deficient and phosphorylation- mimicking) of the GAK phosphorylation site of MYPT1. The phosphorylation- mimicking mutant MYPT1 T443E eliminated the enhanced stress fibre formation in GAK-KO cells (Fig. S5C,D). These findings indicate that GAK activity partially contributes to actomyosin regulation. Therefore, in the original manuscript, we have mentioned that the contribution of GAK kinase activity to GAK-mediated actomyosin regulation is partial (although we have not stated that it does not contribute), but to avoid any misunderstanding that it does not contribute, we have added a statement in the Results section that it does contribute “distinctly” but partially.

3. In a similar way, the authors used stress fibers to test the interplay between GAK and Arhgef2, but this approach is highly unspecific. There is no direct testing of Arhgef2 activity or RhoA activity, which is a significant omission given that the main objective of the study is to understand how GAK inhibits RhoA/ROCK.

Response to Comment 3:

Thank you for your comments. Regarding the interplay between GAK and ARHGEF2, we have previously only evaluated stress fibre formation, but we have now also evaluated MLC phosphorylation (Fig. S6F,G). Furthermore, following the suggestion of another reviewer, we analysed the co-localisation of GAK and ARHGEF2. The results showed that co-localisation with ARHGEF2 was abolished in the delM3 mutant, which lacks the GAK IDR, confirming the importance of the interplay between GAK and ARHGEF2 in actomyosin regulation (Fig. S6C,D). Regarding the direct assessment of RhoA activity, as aforementioned, we were unable to detect RhoA activation throughout the cells (Fig. S7A, B). Therefore, we believe that the direct assessment of transient and highly localised RhoA activity is a challenge that will be addressed more appropriately in future studies. Although we did not directly assess ARHGEF2 activity, the phosphorylation of ARHGEF2 at Ser886 (Ser885 in mice) has been shown to induce its recruitment to microtubules and suppress ARHGEF2 activity (Zenke FT et al. *J Biol Chem.* 2004), and the phosphorylation status of Ser886 was used to evaluate the ARHGEF2 activity (Birkenfeld J et al. *Dev Cell.* 2007; Meiri D et al. *Nat Commun.* 2014). Therefore, we believe that the decreased phosphorylation of ARHGEF2 at Ser 886 in GAK-KO cells is indicative of ARHGEF2 activation (Fig.

7G,H). Furthermore, the fact that the phenotype of GAK-KO cells was attenuated by ARHGEF2 knockdown suggests that ARHGEF2 activity was enhanced in GAK-KO cells (Fig. 7I-L, S6F,G).

4. Given that GAK interacts with additional GEFs (as depicted in Figure 4E), an investigation into the IDR's binding to these other GEFs, beyond just Arhgef2, would be valuable. Although Arhgef2 may have been the initial focus due to a putative GAK phosphorylation motif, the authors' demonstration that the kinase domain is not a primary contributor necessitates exploring whether IDR-mediated inhibition influences the activity of other GEFs and thereby impacts stress fiber formation. Since there is so much binding and a connection to ROCK, it is tempting to speculate that GAK may directly localize to actomyosin filaments, did the author try to localize GAK subcellularly?

Response to Comment 4:

Thank you for your comments. As suggested, regarding GEFs other than ARHGEF2, we also performed additional analyses on ARHGEF1, which showed high co-immunoprecipitation efficiency with GAK, similar to ARHGEF2 (Fig. 4E,F). ARHGEF1 knockdown in GAK-KO cells attenuated stress fibre formation (Fig. S6K,L), and suppressed MLC phosphorylation (Fig. S6I,J), similar to the ARHGEF2 knockdown; however, no significant differences were detected in the cell spread area or aspect ratio (Fig. S6K, M,N). Furthermore, as shown in Fig. S6A,B (original Fig. S3A,B), there was no significant difference in the co-immunoprecipitation efficiency of ARHGEF1 between the IDR-deficient GAK-delM3 mutant and wild-type GAK. These findings suggest that ARHGEF1 is also involved in GAK-mediated actomyosin regulation, but its role may be distinct from that of ARHGEF2, although the details remain unclear.

Indirect immunofluorescence analysis revealed that GAK is primarily localised to the Golgi apparatus in cells, but is also present throughout the cell, including in the nucleus (Fig. S6C,D). In this study using A549 cells, prominent stress fibres were observed in the GAK knockout and knockdown cases, making the co-localisation of stress fibres with GAK difficult to assess.

5. Furthermore, no details are provided regarding the quantitative assessment of stress fibers (methods for segmentation, background subtraction, etc.).

Response to Comment 5:

We appreciate the reviewer's thoughtful suggestion regarding the implementation of actin stress fibre segmentation and background subtraction. We agree that such image processing techniques are beneficial in studies where various types of actin filaments are densely packed together and difficult to distinguish from the background. However, segmentation and background subtraction are not performed. In this experiment, using A549 cells, the baseline phalloidin fluorescence signal was found to be extremely low, and under normal conditions, the cells showed few phalloidin- positive stress fibres. In contrast, the GAK-KO cells showed a significant and clear increase in stress fibre formation. This very low baseline and wide dynamic range provided sufficient separation of the signal from the background without the need for additional segmentation or background subtraction algorithms. In this study, quantification using integrated phalloidin intensity across the whole-cell region yielded consistent results across independent experiments and correlated well with qualitative morphological changes. Implementing advanced segmentation pipelines requires the establishment of new image processing workflows that are beyond the scope of the current study. Therefore, although further improvements could be achieved with a more advanced segmentation- and background-subtraction-based pipeline, we believe that our analysis is sufficient to support our conclusions, and that additional processing would not substantially alter the interpretation of the data. We have clearly stated in the Materials and Methods section that we did not apply segmentation or background subtraction in this study and the reasons for this.

Second decision letter

MS ID#: jcs.264117R1

MS Title: GAK antagonises ROCK-dependent regulation of actomyosin dynamics

Authors: Masaki Hiramoto; Naoharu Takano; Hiroko Kokuba; Hiromi Kazama; Hirotsugu Hino; Hiroshi Handa; Keisuke Miyazawa

Article Type: Research Article

Dear Dr Hiramoto,

We were only able to get one reviewer to look at your revised manuscript. However, the good news is, they were happy with your revisions, so I am happy to tell you that your manuscript has been accepted for publication in Journal of Cell Science, pending standard publication integrity checks.